# Humans and other commonly used model organisms are resistant to cycloheximide-mediated biases in ribosome profiling experiments

Puneet Sharma [1,2,3,5], Jie Wu [1,2,3,4,5], Benedikt S. Nilges [1,2,5] & Sebastian A. Leidel [1,2,3✉]

Ribosome profiling measures genome-wide translation dynamics at sub-codon resolution. Cycloheximide (CHX), a widely used translation inhibitor to arrest ribosomes in these experiments, has been shown to induce biases in yeast, questioning its use. However, whether such biases are present in datasets of other organisms including humans is unknown. Here we compare different CHX-treatment conditions in human cells and yeast in parallel experiments using an optimized protocol. We find that human ribosomes are not susceptible to conformational restrictions by CHX, nor does it distort gene-level measurements of ribosome occupancy, measured decoding speed or the translational ramp. Furthermore, CHX-induced codon-specific biases on ribosome occupancy are not detectable in human cells or other model organisms. This shows that reported biases of CHX are species-specific and that CHX does not affect the outcome of ribosome profiling experiments in most settings. Our findings provide a solid framework to conduct and analyze ribosome profiling experiments.

[1] Max Planck Research Group for RNA Biology, Max Planck Institute for Molecular Biomedicine, Muenster, Germany. [2] Cells-in-Motion Cluster of Excellence, University of Muenster, Muenster, Germany. [3] Research Group for RNA Biochemistry, Department of Chemistry, Biochemistry and Pharmaceutical Sciences, University of Bern, Bern, Switzerland. [4] Graduate School for Cellular and Biomedical Sciences, University of Bern, Bern, Switzerland. [5]These authors contributed equally: Puneet Sharma, Jie Wu, Benedikt S. Nilges. ✉email: sebastian.leidel@unibe.ch

Protein synthesis is a multilayered process involving a plethora of factors that need to be coordinated in response to cellular and environmental cues[1]. The ribosome, an intricate macromolecular machine, is central to this process as it translates the genetic information encoded in messenger RNA (mRNA) into a peptide sequence[2,3]. Even though work from many laboratories has provided detailed insights into the biochemical and kinetic aspects of translation—often using model substrates—technical limitations have long hampered our understanding of its regulation at a global level.

The development of ribosome profiling, a high-throughput method that enables the quantitative analysis of transcriptome-wide translation by determining ribosome positioning at sub-codon resolution[4], has revolutionized the analysis of cellular translation. Ribosome profiling has allowed the characterization of core properties of translation[5], factors governing translational dynamics[6–8], and the analysis of individual translation events[9–13]. The method is based on deep sequencing of ribosome-protected mRNA fragments (RPFs or footprints) that are protected from nucleolytic degradation[4,14]. However, ribosome profiling studies have reached different conclusions on, e.g., the identification of ORFs in zebrafish[9,10,15], the influence of wobble base-pairing on elongation speed[16,17], or the relationship between elongation rates and tRNA abundance[13,18].

These discrepancies are in part caused by adaptations to the original protocols to match the needs of individual laboratories[4,10,19–22]. The differences include the methods of sample harvesting[19,23], nuclease digestion[24], preparation of libraries by circularization or dual linker ligation, and more[4,9,22,25–27]. These small but crucial differences may change the representation of in vivo translational dynamics making direct comparisons between datasets challenging. When harvesting any sample, two key elements likely influence the outcomes of ribosome profiling experiments: The procedure and speed of harvesting, and the use of translation inhibitors. To faithfully capture the in vivo conformation and positioning of translating ribosomes, the time span between the onset of harvesting and flash-freezing of samples is kept at a minimum to prevent ribosome run-off[23]. A second failsafe against undesired elongation in vivo and in the lysate, is the treatment with the translation inhibitor cycloheximide (CHX) used since the beginning in yeast[4] and mammalian cells[28]. CHX is a small molecule that inhibits translation elongation by binding to the ribosomal E-site[29,30]. Several studies, particularly in Saccharomyces cerevisiae, have reported biases in response to CHX treatment[31–36]. CHX appears to reversibly interact with ribosomes, allowing them to move away from their initial position on mRNA[33]. This movement depends on codon identity, therefore, altering the outcomes of studies that require codon-level resolution for occupancy and translation-speed measurements[33]. Furthermore, high concentrations of CHX inhibit elongation, but not initiation[29] leading to ribosome accumulation at the start codon following CHX pre-treatment[5].

Despite those concerns, the addition or omission of CHX has not been systematically compared in experiments using standardized conditions to the best of our knowledge. Most analyses compared published ribosome profiling experiments performed by different laboratories or datasets that varied several conditions simultaneously. Therefore, some conclusions on the effects of translation inhibitors may be affected by parameters other than CHX usage. Finally, the effect of CHX has been predominantly studied in S. cerevisiae and we know little about its impact in vertebrates and other model organisms.

Here, we perform a comprehensive analysis of the effect of CHX on ribosome profiling libraries in human and yeast in parallel experiments. We use a highly standardized and optimized protocol that reduces ribosomal RNA (rRNA) contamination, narrows the footprint distribution, and captures high numbers of ribosome footprints in the dominant reading frame. We found that—if properly handled—the use of CHX does not distort ribosome profiling libraries in human cells. The inhibitor does not affect the quantification of global translation levels, codon-specific ribosome occupancy or the translational ramp. Furthermore, human ribosomes are not susceptible to conformational restrictions by CHX. Similarly, we found that CHX-mediated biases related to rare codons are absent from commonly studied model organisms including two yeast species. These findings show that the effects of CHX are species-specific and do not affect the measurements of translation dynamics in many commonly used model organisms except for baker's yeast. Finally, our results emphasize that the standardization and reporting of parameters beyond the use of CHX will be crucial to improve the comparability of ribosome profiling experiments.

## Results

### Stronger digestion of the cell lysate improves footprint size distribution and frame information

To comprehensively analyze the effect of CHX on human cells, we focused on HEK 293T cells as a well-established model in ribosome profiling. To directly compare our findings to previously reported biases, we performed parallel experiments in S. cerevisiae. Translating ribosomes protect mRNA from nuclease digestion during ribosome profiling experiments, giving rise to two distinct footprint sizes[4,32]. Libraries prepared from footprints excised between ~18 and 32 nt exhibit a bimodal size distribution that centers around 21 nt and 28 nt (in yeast)[4,32] or 21 nt and 30 nt (in mammalian cells)[28,37]. These discrete footprint sizes, 21–22 nt (short footprints) and 28–32 nt (long footprints), have been attributed to distinct conformational states of translating ribosomes[32,37]. If nuclease digestion is stringent, unprotected nucleotides are efficiently trimmed leading to a characteristic three-nucleotide periodicity of mapped reads[4]. The larger the fraction of footprints that map to the correct reading-frame of the ORFs the higher the confidence, with which ribosomal A-, P- and E-sites can be assigned within the footprints. Therefore, we initiated our analysis by establishing a robust protocol that minimizes biases and captures a majority of footprints in the correct reading frame. We harvested and lysed HEK 293T cells and yeast by rapid cryogenic freezing and analyzed the effect of different digestion strengths on library quality by varying RNase I concentration. Since short footprints are sensitive to CHX treatment in yeast, we did not pre-incubate human or yeast samples with the translation inhibitor and lysed the cells in a buffer devoid of the drug.

In human cells a strong RNase I digestion centers footprint-distribution at around 21 and 30 nt (Supplementary Fig. 1a). Importantly, 87% and 75% of the short and long footprints, respectively, were in the correct reading frame in samples treated with 900 U RNase I (Supplementary Fig. 1a, b). A strong digestion of human samples provided the additional benefit of decreasing the amount of rRNA in the libraries (Supplementary Fig. 1c). Interestingly, the amounts of other contaminants upon stronger digestion remain constant but the identity of contaminating species within those fraction changes in the short footprints (Supplementary Fig. 1d). Similar to human cells, yeast samples substantially benefited from stronger RNase I digestion resulting in a pronounced increase in the fraction of in-frame reads for 28 nt footprints when using 600–800 U RNase I (Supplementary Fig. 1e, f). Using our protocol, >60% of the reads uniquely map to the yeast transcriptome without using rRNA depletion. Therefore, we generated ribosome profiling libraries from lysates that were treated with high concentrations of RNase I (900 U for HEK 293T cells and 600 U for yeast).

### Cycloheximide differentially affects human and yeast ribosomes

CHX induced biases have been reported in ribosome profiling data from yeast[31–36]. However, even in yeast, most

analyses compared datasets that were not generated in parallel, and no study has systematically investigated the effects of CHX at the different steps of sample preparation in human cells. Hence, we directly compared three treatments (Fig. 1a and Supplementary Fig. 1g): First, we incubated HEK 293T cells for 1 min in a medium containing CHX prior to lysis and included the drug in the lysis buffer (+/+). Second, cells were subjected to CHX only in the lysis buffer (−/+). Finally, we lysed the cells without CHX pre-incubation in a buffer devoid of the drug (−/−). All experiments were performed in parallel and cells were lysed

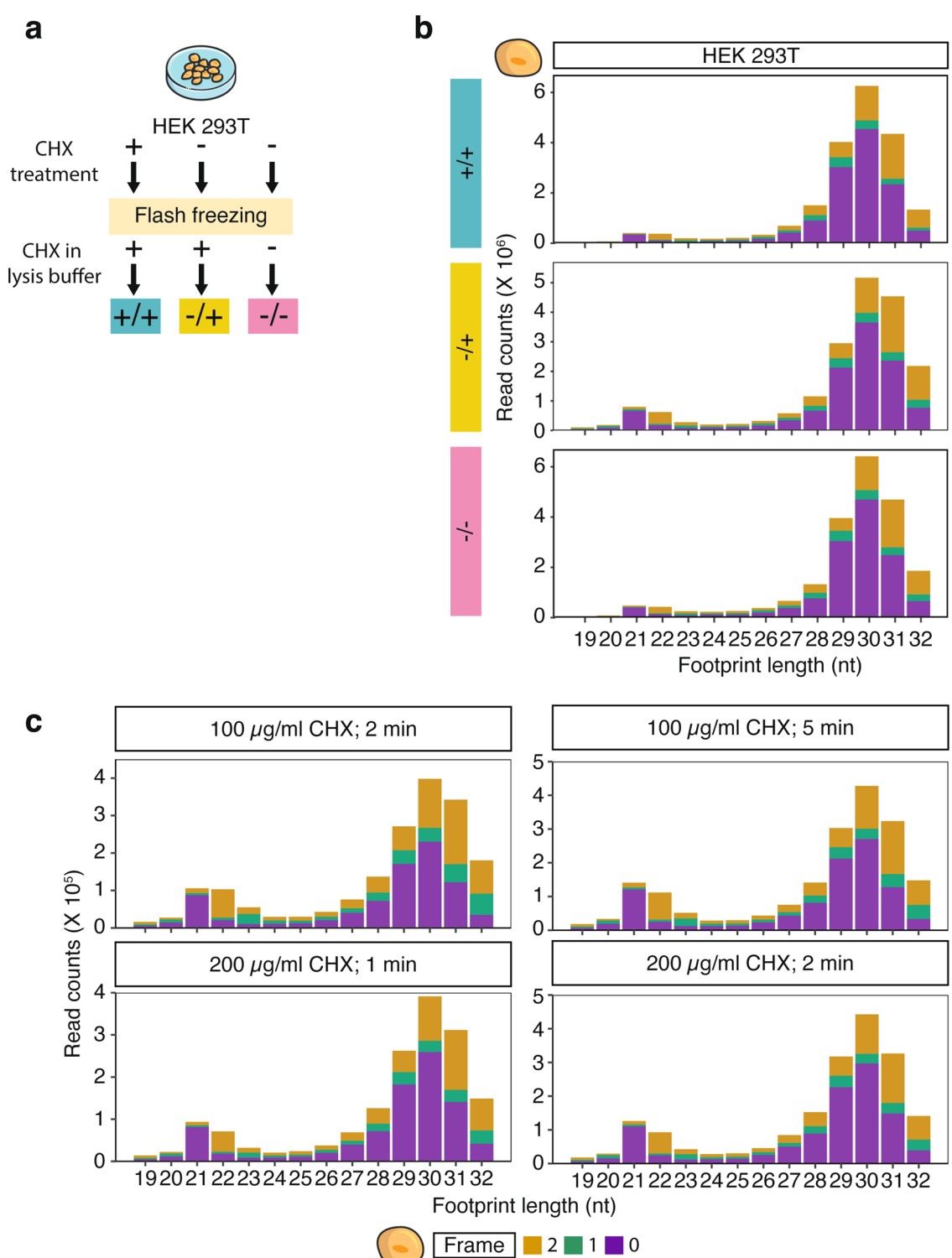

**Fig. 1 Cycloheximide (CHX) does not affect ribosome footprint length distribution in HEK 293T cells. a** Schematic overview of the harvesting and CHX-treatment conditions used for HEK 293T cells. CHX treatment conditions in this and all figures are indicated by color: +/+, green; −/+, yellow; −/−, pink. **b** Representative histograms showing the influence of CHX on footprint length and reading-frame distribution in HEK 293T libraries. The reading frame in all figures is indicated by color: 0, purple; 1, green; 2, yellow. **c** Same as **b** for HEK 293T cells treated with different CHX concentrations and incubation period. Footprints were excised between 18 and 32 nt.

cryogenically in liquid nitrogen to limit ribosome run-off. For inter-species comparisons, we harvested yeast by rapid filtration and flash froze the cells using the same three treatment conditions.

In yeast, CHX stabilizes ribosomes yielding primarily long footprints. Short footprints mainly occur if no CHX is used or if it is applied in combination with inhibitors like tigecycline[32,37]. To examine whether CHX similarly skews the ribosomal population towards large footprints in human cells, we isolated RNase-protected fragments between 18 and 32 nt. Surprisingly, we observed that human ribosomes contain short footprints irrespective of CHX treatment (Fig. 1b). To exclude the possibility that the CHX treatment of HEK 293T cells was too weak to mediate a shift to long footprints, we increased both incubation time and drug concentration. However, the short footprints persisted regardless of the duration or strength of CHX treatment (Fig. 1c). In agreement with our observation, short footprints can even be observed in human cells following extended CHX treatment of up to 24 h[38] (Supplementary Fig. 1h–j) or in combination with tigecycline (Supplementary Fig. 1k–m)[37]. Furthermore, we confirmed the published observations in yeast, where small footprints only occur when CHX is omitted (−/−) but are barely present if CHX is used in the lysis buffer (−/+ and +/+) (Supplementary Fig. 1n)[32,37]. These findings suggest that regardless of the concentration or duration of the CHX treatment or the use of translation inhibitor cocktails, human ribosomes do not completely arrest in only one conformation. These findings further show that CHX affects ribosome profiling experiments in human and yeast differently and that the use of CHX in ribosome profiling has to be assessed for each organism separately.

During the translation elongation cycle, ribosomes adopt several conformations. Nevertheless, only two distinct footprint sizes can be isolated in ribosome profiling experiments[32,37]. Recently, cocktails of translation inhibitors were introduced to enrich for specific steps of translation elongation[37]. In yeast libraries that used inhibitor cocktails short footprints appear to more accurately reflect translation than classical long footprints, while this difference is less striking if only CHX is used[37]. To assess the situation in human cells, we analyzed short and long footprints of the published HeLa datasets[37]. Interestingly, HeLa cells starved for glutamine show stronger codon-specific effects in short footprints due to perturbed charging of tRNA$_{UUG}^{Gln}$ and tRNA$_{CUG}^{Gln}$ (Supplementary Fig. 1o, left). However, the long footprint data showed the same result. While the scale of the effect was weaker, the data were less noisy for long footprints (Supplementary Fig. 1o, right). Since the implementation of ribosome profiling, hundreds of studies have analyzed long footprints. Furthermore, analyses of codon-specific perturbations by using tRNA modification mutants have shown that long footprints provide robust information about translational slowdown[8,39–43]. Therefore, we focused our analyses on long footprints and included short footprints for comparison whenever possible.

**Translation ramp and ribosomal occupancy are independent of CHX**. Ribosome profiling data can be used to infer elongation rates along a transcript, because footprint densities reflect the local residence times of ribosomes[13]. Several studies in yeast have described an increase in ribosome density in the first ~200 codons of ORFs. This phenomenon is called the "5′ translation ramp" and is thought to be caused by a low elongation speed in this region[7,44,45]. In contrast, a study in murine embryonic stem cells concluded that a similar ramp does not occur in vertebrate cells[5]. To elucidate the effect of CHX on this phenomenon, we analyzed footprint density at the beginning of the open reading frames. Importantly, we found an enrichment of ribosomes from 15 to ~150 codons irrespective of drug treatment and footprint length in human cells (Fig. 2a and

Supplementary Fig. 2a–c). The scale of this effect and the position of the maximum footprint density differ between human and yeast (~60 codons in humans, and ~40 codons in yeast; Supplementary Fig. 2d). However, the ramp is independent of gene-translation levels and gene length (Supplementary Fig. 2b, c, e). This shows that the translation ramp itself is not induced by CHX but is a genuine biological feature of translation in humans as well as in yeast. Multiple factors were proposed to trigger the translational ramp, such as the presence of rare codons with low cognate tRNA availability, mRNA secondary structures, or the interaction of nascent polypeptides with the ribosomal exit tunnel[39,40,46]. Our results exclude that CHX treatment, gene length, or transcript expression levels are primary contributing factors (Fig. 2 and Supplementary Fig. 2). However, ribosome density is increased in the first ~15 codons in human and yeast +/+ libraries and to a lesser extent in the −/+ and −/− libraries (Fig. 2a and Supplementary Fig. 2a, b). This is consistent with the observation that CHX arrests elongation but does not block translation initiation at the concentration generally used in ribosome profiling[29]. Therefore, we do not recommend to pre-treat cultures with CHX. In case that an experiment requires CHX pre-incubation, time should be kept to a minimum. Furthermore, the first codons should be excluded from gene-expression analyses since this accumulation will skew the analysis of differential gene translation towards initiation rates and does not reflect global gene translation levels.

Our analysis of the translation ramp suggested, that CHX does not induce a global effect on ribosome coverage in human cells. To similarly confirm this at the level of gene expression, we counted uniquely mapped ribosome footprints across the human transcriptome and determined differential abundance between +/+, −/+, and −/− libraries using DESeq2 (Fig. 2b and Supplementary Data 1)[47]. We observed very few transcripts (<24 out of 12,330 analyzed) that were significantly altered by CHX treatment (Fig. 2b; blue dots). The absence of deviations in ribosome occupancy upon CHX treatment was not caused by technical variability of the sequencing libraries since many transcripts were unaffected (long footprints, 46–48%) with statistical certainty (Fig. 2b; top right; red dots). For short footprints, only 14% of the transcripts are unaffected with statistical certainty due to the low read count for many genes and strict filtering of genes with low counts during differential expression analysis with DESeq2 (Fig. 2b; bottom left; red dots). Thus, human cells do not display altered ribosome occupancy at the transcript level upon CHX treatment, and translation levels can be compared irrespective of CHX use.

Next, we analyzed footprint density around the annotated start and stop codons. The enrichment of reads at the initiation site in response to CHX pre-treatment can be exploited to identify upstream open reading frames (uORFs) or other features that are associated with initiation. Recently, it was reported that translation initiation at non-AUG start codons encoding for uORFs or N-terminal protein extensions is resistant to CHX treatment[38]. Therefore, we tested whether the different treatment conditions influence ribosome occupancy in the 5′ untranslated region (UTR) and compared the reads mapping to the region upstream of annotated AUG start codons. Using long footprints, we found 317 upregulated and 257 downregulated 5′ UTRs in the +/+ condition compared to −/+ (Supplementary Fig. 2f). In contrast, comparing −/− and -/+ libraries revealed only 7 downregulated UTRs. For short footprints, the low number of UTR reads did not allow for a meaningful comparison. Thus, CHX pre-treatment alters the footprint density in the 5′ UTR and might be an interesting strategy to complement drugs like harringtonine or lactimidomycin for uORF identification[5,20].

Finally, we observed an accumulation of ribosomal footprints at stop codons in all three conditions, but most prominently in

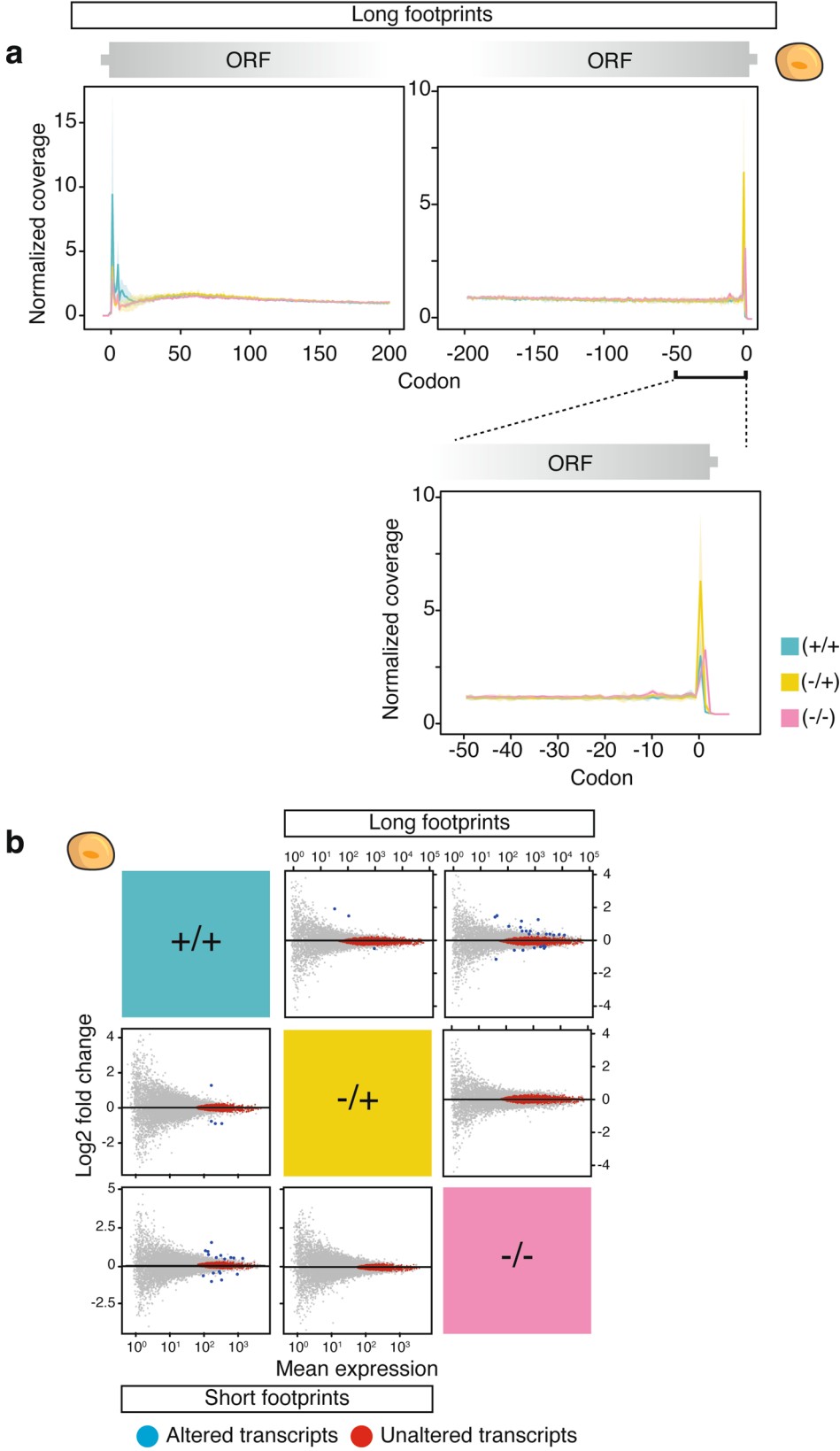

**Fig. 2 Cycloheximide (CHX) does not affect the translation ramp or global translation levels. a** Normalized ribosomal A-site coverage observed in long footprints (29–31 nt) for the first 200 (left) and last 200 (right) codons in HEK 293T cells in highly expressed genes (>64 reads). The solid line depicts the mean and shaded areas represent 95% confidence intervals for three biological replicates ($n = 3$). **b** Differential ribosome occupancy of open reading frames (ORFs) in HEK 293T cells across inhibitor treatments were identified using DESeq2, which uses a negative binomial distribution model with fitted mean[47]. Uniquely mapped reads were split into short footprints (bottom left; 21–22 nt) and long footprints (top right; 28–32 nt). ORFs, after excluding the first 15 codons, were tested for differential translation (adjusted $p$-value $\leq 0.05$) and for unaltered translation (adjusted $p$-value $\leq 0.05$). Significantly altered ORFs are indicated in blue, unaltered transcripts in red.

the −/+ libraries (Fig. 2a and Supplementary Fig. 2a, d), consistent with reports in murine and yeast cells[5,33,45]. This emphasizes that the steps during translation termination differ from normal translation elongation and is consistent with the observation that termination complexes are differentially stabilized by the addition of CHX[48]. It is possible that ribosome splitting is affected by CHX and does not occur in the −/+ extracts while it can still occur in untreated −/− samples. While it is yet unclear, which of the three treatment options captures the in vivo situation at the stop codon most realistically, this effect can likely be exploited to complement in vitro termination experiments.

**Cycloheximide does not affect A-site codon-level ribosome occupancy in human cells.** CHX arrests translation by binding to the ribosomal E-site thereby locking ribosomes in the pre-translocation state[37]. However, CHX-binding kinetics might differ for specific codons present in the ribosomal sites leading to their mis-representation as described for CGA and—albeit weaker—CGG in yeast[33]. This would be perceived as an alteration in the decoding kinetics of specific codons but may remain undetected when quantifying occupancy across transcripts consisting of hundreds of codons. To test for the CHX-dependent enrichment of codons, we calculated transcriptome-wide codon occupancy for the A-, P-, and E-site codons in human and yeast (Fig. 3a–c and Supplementary Fig. 3a). For short and long footprints, correlations of A-site-codon occupancy in HEK 293T libraries were very high, independent of the treatment ($R^2 \geq 0.92$ between all conditions; Fig. 3a). Similarly, P-site-codon occupancy correlated well between treatments ($R^2 \geq 0.74$; Fig. 3b) while CHX pre-treatment changed E-site-codon occupancy markedly ($R^2 = 0.28$–$0.35$ for +/+ and −/− conditions; Fig. 3c). A-site codon occupancy in human cells was not strongly affected by the digestion strength during footprint generation (Supplementary Fig. 3b). Similarly, the duration of CHX incubation or the use of a different cell line (HeLa instead of HEK 293T) only played a minor role (Supplementary Fig. 3c). This shows that codon-specific analyses except for the E-site can be conducted in human cells independent of CHX treatment. In contrast to human cells, pre-incubation with CHX was the key factor that affected A-site-codon correlations between yeast samples ($R^2$ +/+ vs. −/+ = 0.04 and $R^2$ +/+ vs. −/− = 0.15; Supplementary Fig. 3a), while the addition of the inhibitor to the lysis buffer had a low impact on A-site-codon occupancy ($R^2$ −/− vs. −/+ = 0.86; Supplementary Fig. 3d). This further emphasizes that the use of CHX in ribosome profiling needs to be viewed in a species-specific manner and that its use in human cells does not distort the data even at the codon level when applied carefully.

To estimate the influence of CHX at the codon level in yeast, we compared A-site-codon occupancy in a large number of libraries that used different CHX-treatment conditions in wild-type yeast[8,31,32,49–51]. It is striking that +/+ experiments correlate well between laboratories, but not with −/+ or −/− (Supplementary Fig. 3d). However, it is difficult to assess which type of treatment reflects the in vivo situation best. To address this question, we analyzed 5′P sequencing (5PSeq) and translation complex profile (TCP) sequencing libraries, which use orthogonal strategies to determine ribosomal positions across the transcriptome[52,53]. 5PSeq analyses exonucleolytically degraded 5′-monophosphorylated mRNA intermediates, thereby determining the position of the 5′-most ribosome on a mRNA that undergoes degradation[52]. TCP-seq uses formaldehyde fixation of living cells in the absence of CHX prior to the generation of ribosome profiling libraries allowing the identification of ribosomal complexes that undergo scanning and initiation[53].

Interestingly, results of TCP-seq correlate well with −/+ and −/− libraries ($R^2$ −/+ vs. TCP-seq = 0.69; $R^2$ −/− vs. TCP-seq = 0.50), but not with +/+ libraries ($R^2$ +/+ vs. TCP-seq = −0.21; Supplementary Fig. 3e, f). Similarly, 5PSeq correlates more with the libraries that do not use CHX pre-incubation (Supplementary Fig. 3e, f). In summary, the analyses of TCP-seq and 5PSeq support the idea that the pre-incubation with CHX is the critical difference between libraries and that samples without pre-incubation are most suitable to represent the in vivo situation in baker's yeast.

**The absence of CHX-dependent ribosomal waves is consistent across most species.** One of the main reasons behind contradictory translation rates of specific codons in yeast was the observation of increased ribosome occupancy downstream of specific codons in CHX-treated samples[33]. These ribosomal "waves" occur downstream of the rare CGA and CGG codons. CGG is decoded by the essential $tRNA_{CCG}^{Arg}$, which is expressed from a single gene in baker's yeast[54,55]. CGA exhibits a more pronounced wave than CGG and is read by $tRNA_{ICG}^{Arg}$ requiring a wobble interaction to be decoded[33,56]. As expected, we observed a wave for CGA and CGG in the +/+ but not in the −/+ and −/− libraries in yeast (Fig. 4a). However, we did not observe a similar effect for CGA in HEK 293T cells (Fig. 4b and Supplementary Fig. 4). In humans, decoding of CGA relies on its cognate $tRNA_{UCG}^{Arg}$ and does not require wobble interactions[54,55]. Furthermore, there are no codons in humans that are decoded by a single tRNA like $tRNA_{CCG}^{Arg}$ in yeast (Supplementary Table 1). Consistently, we did not observe wave formation for any codon in humans. Even for the UUA codon that is both rare and has a low tRNA copy number in humans, we did not detect CHX-induced downstream enrichment of ribosomes (Fig. 4b and Supplementary Fig. 4). These ribosomal waves are not only absent from our human-cell libraries, but from all published human ribosome profiling datasets that we analyzed (Fig. 4c). Interestingly, this effect is not specific to human cells, because ribosomal waves are absent from ribosome profiling data of other vertebrates (Fig. 4a). Neither zebrafish[10] nor mouse[5] show ribosome enrichment downstream of any codon. Finally, to investigate whether altered translation at rare codons is a general feature of yeasts, we generated ribosome profiling libraries from CHX-treated Schizosaccharomyces pombe and Candida albicans. Like in S. cerevisiae, CGG is decoded by a single copy of $tRNA_{CCG}^{Arg}$ in both species. However, while CGA is decoded by its cognate $tRNA_{UCG}^{Arg}$ in S. pombe, it depends on decoding by $tRNA_{ICG}^{Arg}$ in C. albicans. Surprisingly, we did not observe ribosomal waves at CGA, CGG or any other codon in both yeasts (Fig. 4d). This shows that ribosomal waves are specific to baker's yeast and that the use of CHX is, therefore, mainly a concern when studying translation in S. cerevisiae, but not in other widely used eukaryotic model organisms.

**Variability in library preparation or culture conditions may contribute to the poor correlation between similar samples.** We found that the impact of cycloheximide depends on the model organism. Furthermore, the use of CHX in the lysis buffer did not affect the correlation of A- and P-site codons in human cells. Therefore, we analyzed how much variability exists between ribosome profiling datasets generated by different laboratories. Surprisingly, ribosome profiling data from different laboratories differed substantially, independent of whether HEK 293 or HEK 293T cells were used (Fig. 5a). This is less striking for yeast samples, where the difference between datasets is mainly explained by CHX pre-incubation (Supplementary Fig. 3d). When comparing the protocols

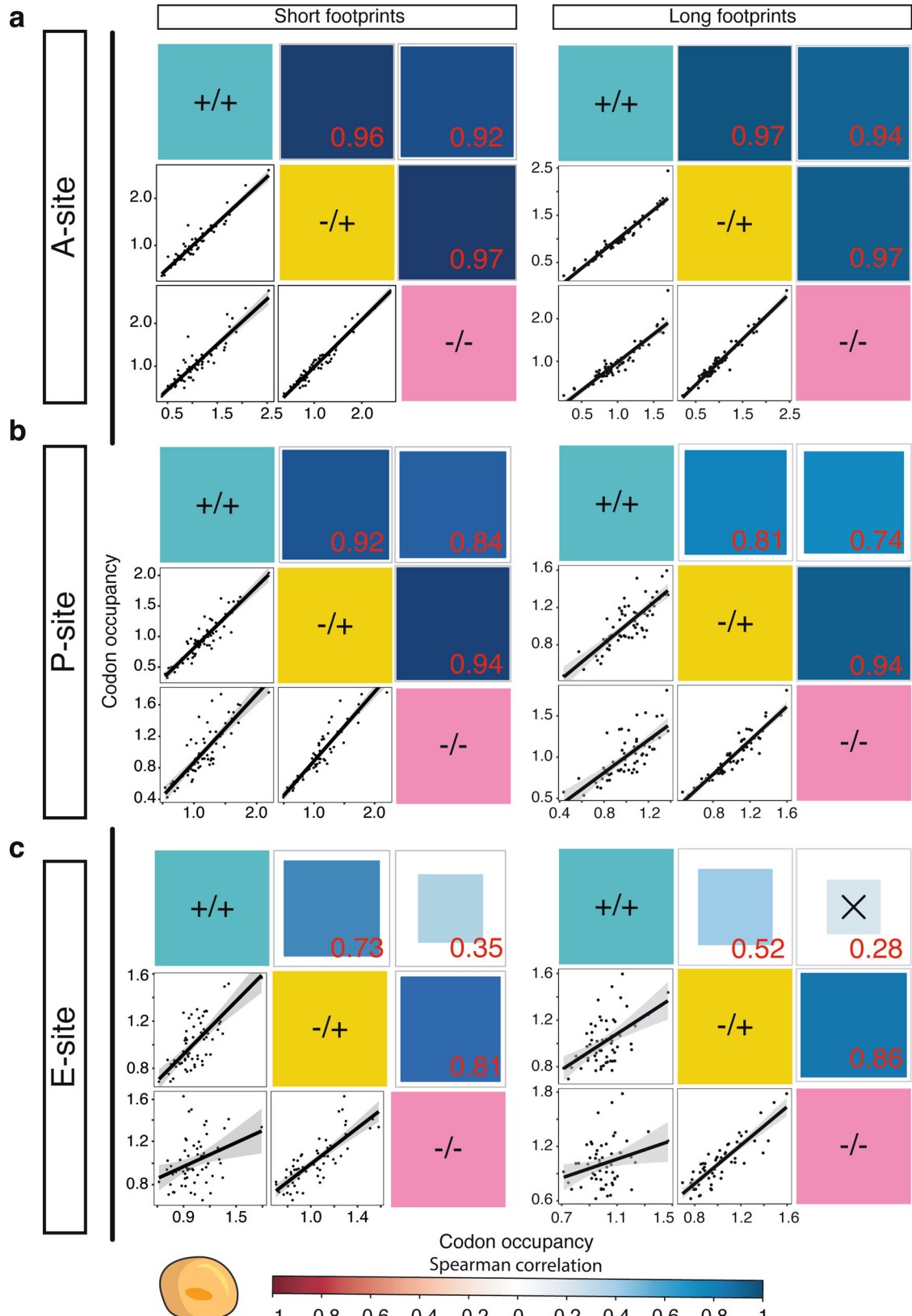

**Fig. 3 Cycloheximide (CHX) does not alter mammalian ribosome occupancy at the A-site and the P-site. a** Correlation analysis of transcriptome-wide A-site codon occupancy in HEK 293T cells across different CHX treatments for short and long footprints. The solid line depicts the fitted line and shaded areas represent 95% confidence intervals for three biological replicates ($n = 3$). Each black dot represents a codon. The size of the box indicates $p$-values. Correlations with a $p$-value > 0.05 are crossed out. **b** Like **a** for P-site codon occupancy. **c** Like **a** for E-site codon occupancy. Footprints were excised between 18 and 32 nt but are represented according to size: short footprints (21–22 nt) and long footprints (29–31 nt).

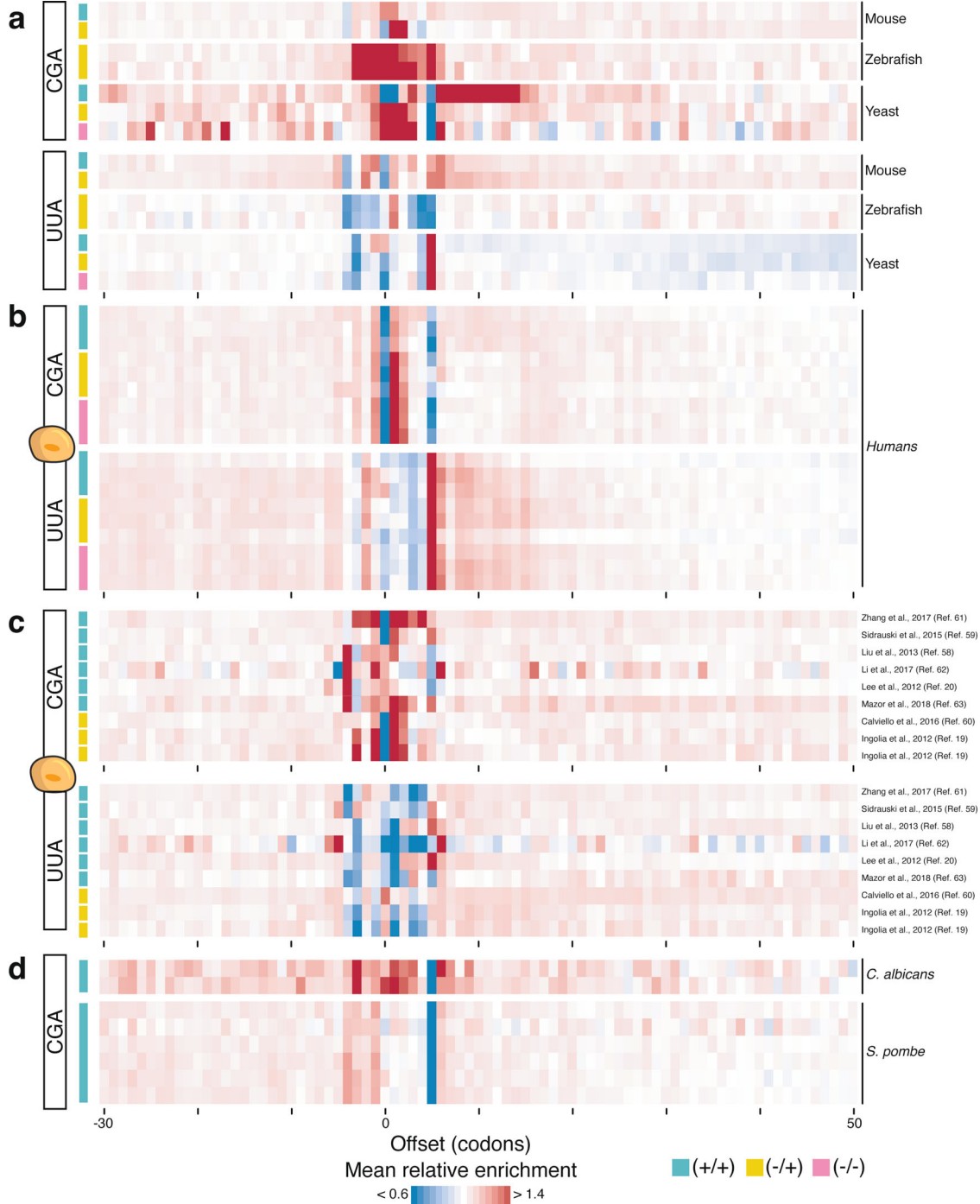

**Fig. 4 Cycloheximide (CHX) pre-treatment does not alter ribosome occupancy downstream of rare codons in most species. a** Transcriptome-wide ribosome enrichment profiles according to Hussmann et al.[33] using libraries generated from E14 mouse embryonic stem cells[5], zebrafish embryos[10], and wild-type yeast (this study) surrounding CGA and UUA codons and using different CHX treatment regimens. **b** Same as **a** for long footprints of HEK 293T cells. **c** Same as **a** for published human ribosome profiling datasets[19, 20, 58-63]. **d** Same as **a** for the CGA codon in *C. albicans* and *S. pombe* (this study). This plot only uses long footprints (28–32 nt).

carefully, we found a large variability in cell harvesting, gradient centrifugation, the amount of cell lysate used for digestion, RNase I digestion temperature, digestion time, and the library-preparation methods (Supplementary Table 2). Furthermore, passage number, the time between feeding cells, and harvesting or media age were factors that we were unable to assess. These seemingly minor details are often overlooked when comparing ribosome profiling experiments but appear to have profound effects on library characteristics (Supplementary Fig. 5a, b). We excluded that either the strength of

digestion, the cell line, or CHX incubation time are key factors that can individually explain the observed differences in A-site codon occupancy (Supplementary Fig. 3b, c). Furthermore, we analyzed the genetic diversity of the cells based on differences in single-nucleotide polymorphisms (SNPs). However, we did not find evidence that the cells differed significantly. Therefore, our analysis suggests that a combination of factors causes these differences and that the influence of these steps in the protocol is more relevant than generally thought. In human cells, these subtle differences can outweigh the

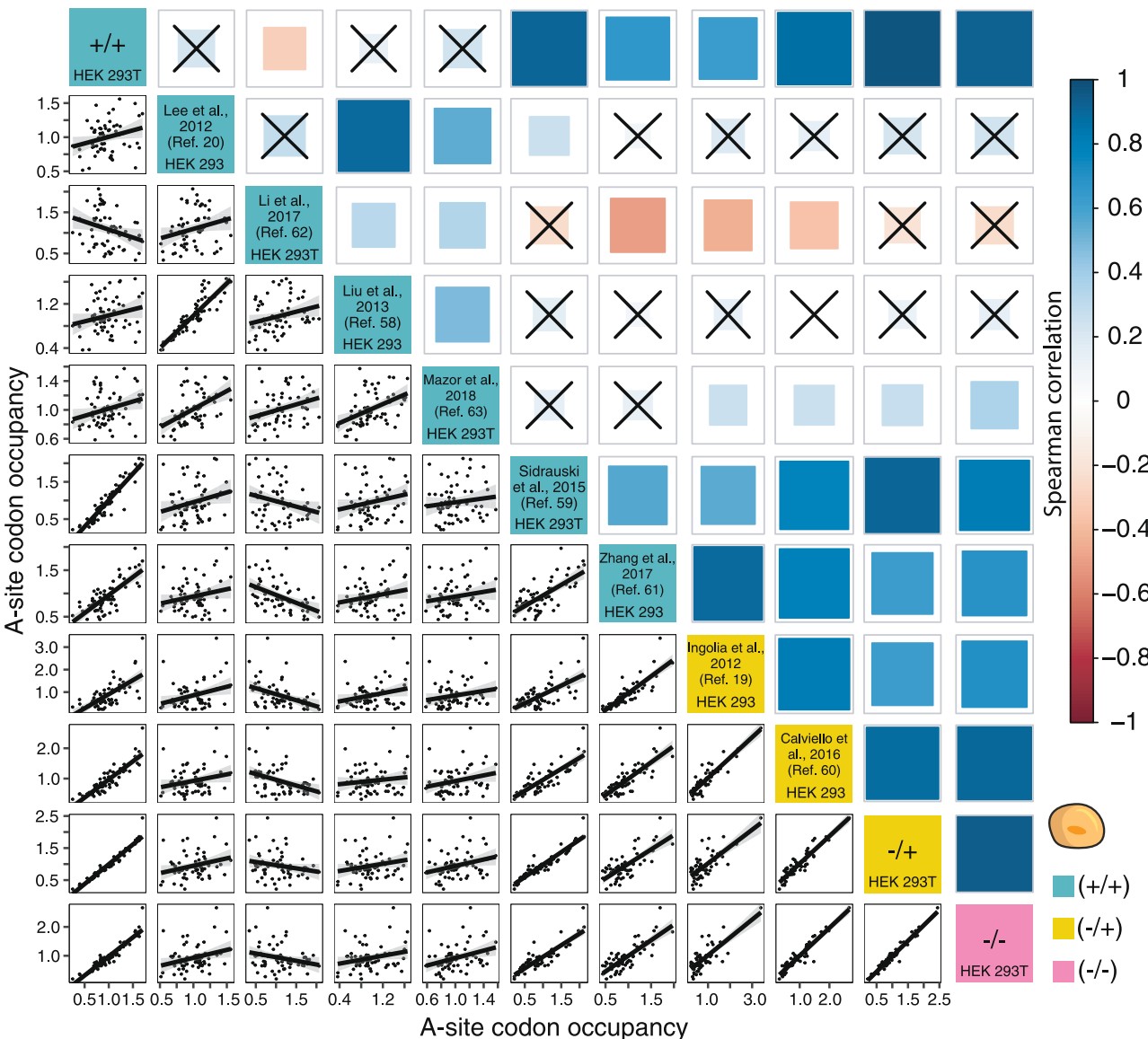

**Fig. 5 Cycloheximide (CHX) does not explain the poor correlation between various datasets in humans.** Correlation analysis of transcriptome-wide A-site codon occupancy across data from this study and published datasets for human cells (HEK 293 and HEK 293T cells) using different CHX treatments[19, 20, 58–63]. The solid line depicts the fitted line and shaded areas represent 95% confidence intervals for three biological replicates ($n = 3$). Each black dot represents a codon. The size of the box indicates p-value. Correlations with a p-value > 0.05 are crossed out.

effects of CHX treatment since libraries prepared by the same laboratory often correlate better despite inhibitor use when compared to samples from different laboratories using the same experimental regimen.

## Discussion
Since its implementation more than a decade ago ribosome profiling has undergone continuous adaptations and improvements. Even though some aspects of the protocol—like the treatment with CHX—have been the focus of intense discussions, other aspects have received less attention. We found that stronger nuclease digestion improves reading-frame information and narrows read-length distribution in both human and yeast. Furthermore, this strategy reduced rRNA contamination (Supplementary Fig. 1c) and enhanced data quality allowing us to better investigate the effect of CHX. To further reduce library contaminations, it might be beneficial to compare cocktails of nucleases and to optimize the RNA-to-RNase ratio[24]. However,

nucleases need to be titrated for each cell line as different cell types contain different amounts of RNase inhibitors[57].

The use of CHX in ribosome profiling has been challenged by reports about CHX-induced biases in baker's yeast[31–36]. This may have even prompted researchers to repeat experiments without the inhibitor to confirm previous findings leading to a duplication of efforts. Therefore, we have conducted a systematic analysis to determine the impact of CHX at the different stages of library preparation in human cells and yeast in parallel experiments. Importantly, we find that the effect of CHX depends on the species. Human ribosome profiling libraries generated by rapid lysis and without prolonged pre-incubation with CHX are devoid of reported CHX-mediated biases. We did not find significant differences in global translation levels, the translational ramp or codon enrichment, findings that we verified by analyzing published datasets[19,20,58–63].

A codon-specific alteration of ribosome occupancy leading to ribosomal waves is observed for CGA and CGG in CHX pre-treated baker's yeast[33]. These codons, which are rare in *S. cerevisiae*, *S.*

pombe, and *C. albicans*, exhibit either low levels of cognate tRNA or depend on wobble decoding[56]. Hence, their tRNA adaptation index (tAI) values, a measure of codon-anticodon decoding that takes into account tRNA copy numbers and wobble-pairing constraints[64], are very low. In vertebrates, CGA is not a rare codon and its tAI values and elongation rate are intermediate. Furthermore, in common vertebrate models, no codon features a similarly low codon frequency and depends on wobble decoding like CGA in yeast, suggesting that these organisms are less likely to be affected by CHX treatment. Consistently, we found no evidence of ribosome enrichment downstream of any codon in zebrafish, mice, and humans. It would require more replicates and deeper sequencing to detect subtle effects in specific ORFs or for specific codon pairs. In baker's yeast, the inefficient decoding of CGA largely depends on wobble pairing to $tRNA_{ICG}^{Arg}$ with a minor contribution of tRNA abundance[56]. However, neither low codon frequency, low tRNA copy numbers nor wobble codon-anticodon interactions can explain CHX-mediated waves as single factors as shown by the comparison with *S. pombe* and *C. albicans* (Fig. 4d). It cannot be excluded that differences in elongation rate or a rate-limiting elongation step that varies between yeast and mammals lead to the observed differences. However, the comparison to *S. pombe* and *C. albicans* makes this less likely. Baker's yeast may vary from other species by its uptake and turnover of CHX or by subtle structural features of the yeast ribosome and how CHX interacts with it. We excluded slower uptake of CHX in human cells as a contributing factor by increasing CHX concentrations and incubation time (Fig. 1c). A recent study reported that the binding site of CHX in human ribosomes is similar to yeast, but that the ligand adopts a different conformation[65]. This might facilitate a more stable binding of CHX to human ribosomes leading to a better preservation of their position on mRNA. Interestingly, a *rpl1bΔ* mutant was shown to improve translation of CGA codons pointing towards structural constraints of yeast ribosomes[66]. Therefore, special care will be required when analyzing CGA. However, this codon similarly triggers phenotypes in orthogonal assays[56,67,68]. Hence, the observed effects likely indicate a true biological feature. To disentangle the molecular mechanisms will necessitate a detailed analysis of CHX-binding to ribosomes in more organisms.

But which CHX treatment should be used for ribosome profiling? In human cells, we did not observe global differences in ribosome occupancy between the treatments. Furthermore, A-site and P-site occupancy are not altered (Fig. 3a, b). To clarify the situation in *S. cerevisiae*, we used data from 5PSeq and TCP-seq to identify the regimen of CHX usage that best matches the in vivo situation. The results speak against CHX pre-treatment (Supplementary Fig. 3e, f). However, the small number of published TCP-seq and 5PSeq libraries did not allow us to distinguish between the −/+ and the −/− conditions. More independent experiments will be required to reach a final conclusion. Interestingly, the two 5PSeq libraries correlated most strongly with each other independent of CHX use (Supplementary Fig. 3e), suggesting that the characterization of the last translating ribosome during mRNA degradation is influenced by factors like the processivity of the degradation machinery in addition to translation dynamics. Therefore, 5PSeq may not be optimal to assess the influence of CHX on ribosome profiling. We conclude that using CHX in the lysis buffer without pre-treatment of the culture has only a minor influence on ribosomal A-site-codon occupancy even in yeast. Therefore, we generally recommend the use of the −/+ protocol for human cells and other species including yeast. Adding CHX to the lysis buffer provides the advantage of limiting ribosomal run-off. If the experimental design requires CHX pre-incubation, the exposure of the cells should be kept as brief as possible (≤1 min) to avoid an accumulation of reads around the start unless this is intended.

We performed our experiments under rich-media conditions in cells that were not exposed to stress during culture and harvesting. We cannot exclude that CHX-mediated effects occur under specific stress and in certain species. The step that appears most vulnerable is initiation (Fig. 2a and Supplementary Fig. 2a–e). However, stresses that affect translation initiation will very likely coincide with a general stress response that will be visible in the gene-expression pattern (e.g., by induction of *GCN4* in yeast)[39,69]. In particular, in light of reports that found changes in translation efficiency in response to starvation upon long pre-treatment with CHX, it is advisable to verify findings by omitting CHX from experiments[36]. However, at this point, there is no evidence of the negative effects of using CHX in the buffer (−/+).

There are three conditions that require specific attention when using cycloheximide: first, E-site codon occupancy is affected by pre-treatment with CHX in human cells. Generally, the E-site is not considered critical for decoding or peptide-bond formation. However, for researchers interested in E-site biology, the use of CHX is not recommended. Since CHX binds to the E-site its presence likely alters the E-site interaction with tRNA thus affecting tRNA release. Second, CHX blocks elongation but not initiation. This leads to an enrichment of reads around the initiation site as a function of the initiation rate and the extent of CHX treatment. While this effect can be used intentionally to enrich for initiation sites in species where harringtonine and lactimidomycin do not work, it is generally advised to exclude the first few codons of a gene from most analyses. Third, termination differs from normal elongation cycles. Ribosomes spend more time at stop codons, which is apparent in all treatment conditions. However, the strongest enrichment is seen in −/+ libraries. Ribosomes are likely able to enter the termination cycle but fail to terminate since CHX in the lysis buffer locks the ribosomes in the unrotated state thereby stabilizing termination complexes[48]. To specifically analyze termination, it might be beneficial to use the different treatments side by side.

Recently, short footprints have been used as a tool to obtain more information about ribosomal conformations in addition to the canonical long footprints[32,37]. In yeast, short footprints show a high anti-correlation with the tAI. However, no significant correlation with 1/tAI is seen in human cells (Supplementary Fig. 5c). Similarly, the correlation between codon occupancy and amino acid polarity observed for short footprints in yeast[32] is absent in human cells (Supplementary Fig. 5d). In fact, we did not find a significant correlation between codon occupancy and most of the biochemical properties of amino acids for both long and short footprints (Supplementary Fig. 5d). Short footprints either derive from rotated ribosomes or from unrotated pre-accommodation-state ribosomes with an empty A-site[32,37]. In libraries that are devoid of CHX short and long footprints simultaneously occur in yeast[32], while both types of footprints are present in human libraries independent of the CHX regimen (Fig. 1b and Supplementary Fig. 1h–m). While empty A-sites can be induced by treatments like specific amino acid starvation or the use of tRNA toxins[37], it is difficult to distinguish these two types of short footprints in vivo. By combining CHX with other inhibitors like tigecycline or anisomycin in yeast specific types of short footprints can be enriched, allowing to further improve the resolution of ribosome profiling[37]. Nevertheless, the use of canonical long footprints and CHX have allowed to identify biologically meaningful features in ribosome profiling experiments, e.g., in the analysis of tRNA modification mutants[8,39–41,43]. We have analyzed several features in ribosome profiling data in response to CHX exposure independently for small and large footprints and found the results to be very similar. Even though subtle differences may exist, these appear to be rather of quantitative than qualitative nature. Currently, it is

unclear whether the simultaneous purification of both footprints sizes increases the information gain in cases when inhibitor cocktails are not used. The recovery of short and long footprints requires a wide size selection from the acrylamide gel, thereby increasing the contamination with rRNA. Since the quality of the results correlates with the amount of usable data an increase in contamination is a factor that needs to be carefully balanced. Thus, unless the research question necessitates the preparation of small footprints, e.g., when empty A-sites are expected, the use of large footprints appears fully justified. This strategy has the added advantage of being able to compare the data to hundreds of published datasets that have used long footprints. Nevertheless, the use of inhibitor cocktails has provided us with new strategies to further probe translation dynamics.

CHX-induced effects have drawn a lot of attention. However, the contribution of other factors has likely been underestimated. While analyzing ribosome profiling data from various studies, we observed discrepancies in library properties such as fragment length, the extent of frame information, and the level of contamination by rRNA or small RNAs (Supplementary Fig. 5a and b). Such differences likely stem from laboratory-specific procedures during, e.g., cell handling, lysis, digestion, or size selection. However, these effects may affect the analysis of translation dynamics more than the use of CHX (Fig. 5) or even the use of different cell lines (Supplementary Fig. 3c). And similar to the use of CHX such effects can be species-specific. For instance, rapid filtration and flash-freezing leads to minimal biases in yeast and have become the gold standard, because harvesting by centrifugation elicits an immediate starvation response[70]. However, the same strategy induces pauses at serine and glycine codons in the A-site of *E. coli*[71].

To ensure meaningful analyses it will be critical to be as transparent as possible about experimental conditions. Since it is difficult to identify the sources of variability, we urge the community to standardize the ribosome profiling protocol and to set clear standards for reporting how libraries are prepared. This includes (I) the duration and exact use of inhibitors, (II) the method of lysis, (III) the concentration of nucleases relative to the amount of nucleic acids, (IV) the duration and temperature of footprinting, and (V) which footprint sizes were selected for library preparation and whether they were pooled or not. Knowing these details for individual experiments will allow the community to select, which published datasets and translational features can be directly compared.

## Methods
A detailed ribosome profiling protocol is provided in the Supplementary Methods.

**Yeast harvesting and footprinting**. Overnight cultures of wild-type yeasts in the BY4741 (*S. cerevisiae*), 972(h⁻) (*S. pombe*), and SN87 (*C. albicans*) backgrounds were diluted and grown to mid-exponential phase (OD$_{600}$ ~ 0.4). For +/+ samples, CHX was added to a total concentration of 100 μg/ml, and cultures were gently agitated for 1 min at 30 °C. Cells were rapidly harvested by vacuum filtration through a 0.45 μm cellulose nitrate filter (GE Healthcare) and immediately flash-frozen. Samples were mechanically lysed under cryogenic conditions in a Freezer-Mill (SPEX SamplePrep) with 2 cycles at 5 CPS interspersed by 2 min of cooling. Lysates were thawed in lysis buffer (20 mM Tris-HCl pH 7.4, 5 mM MgCl$_2$, 100 mM NaCl, 1% Triton, 2 mM DTT) containing 100 μg/ml CHX for +/+ and −/+ samples and clarified by two rounds of centrifugation (5 min; 4 °C; 10,000×$g$). Unless specified otherwise, 10 A$_{260}$ units of cleared lysates were digested with 600 U Ambion RNase I (ThermoFisher) for 1 h at 22 °C and the reaction was inhibited by the addition of 15 μl SuperaseIn (ThermoFisher). Monosomes were isolated from the sucrose gradients according to ref. [8]. All libraries were prepared from footprints in the range of 18–32 nt (HEK 293T) and 18–30 nt (*S. cerevisiae*). These footprints were cut from acrylamide gels using RNA size markers of 18 nt and 30 nt (yeast) or 18 nt and 32 nt (HEK 293T). Finally, libraries were generated using 3′-adapters that were randomized at the first 4 positions of the 5′ end to minimize potential ligation biases[19,22]. A detailed protocol can be accessed in supplementary methods.

**HEK 293T cell harvesting and footprinting**. For the +/+ samples, cultured HEK 293T cells were incubated with a medium containing 100 μg/ml CHX for 1 min, washed with ice-cold PBS, and flash-frozen in liquid nitrogen. The dish was swiftly transferred to ice and 400 μl lysis buffer (10 mM Tris-HCl pH 7.5, 100 mM NaCl, 5 mM MgCl$_2$, 1% Triton X-100, 0.5 mM DTT, 0.5% deoxycholate (w/v) and 100 μg/ml CHX) was dripped onto the cells. The cells were harvested on ice by scraping once the lysis buffer was thawed. Cells for −/+ and −/− conditions were harvested like +/+, however, CHX pre-incubation was omitted and it was not added to the lysis buffer for −/−. Samples were clarified by centrifugation (5 min; 4 °C; 10,000×$g$). Unless specified otherwise, 10 A$_{260}$ units of cleared lysates were digested with 900 U RNase I (ThermoFisher) for 1 h at 22 °C and the reaction was inhibited by the addition of 15 μl SuperaseIn (ThermoFisher). Monosomes were isolated from the sucrose gradients and libraries were prepared analogous to yeast samples.

**Sequencing, processing, and mapping of reads**. Ribosome profiling and RNA-Seq libraries were sequenced on Illumina HiScanSQ and NextSeq sequencers. Ribosome profiling reads were processed by clipping the adapter sequence and trimming the 4 randomized nucleotides of the linker using the FASTX-Toolkit (http://hannonlab.cshl.edu/fastx_toolkit, June 2017), version 0.0.13. Processed yeast reads were mapped to tRNA, rRNA, snRNA, and snoRNA genes from SGD to remove possible contaminants using bowtie[72] version 1.0.0. Processed reads from human samples were mapped to rRNA using bowtie version 1.2.1.1. Residual reads were uniquely mapped to non-dubious ORFs (sgdGene) using bowtie. Reference ORFs (hg38 UCSC canonical transcripts; mm10 UCSC genes; danRer10; C_albicans_SC5314_version_A22; Pombe_ASM294v2) were extended by 18 nt into the UTRs to allow alignment of footprints from initiating and terminating ribosomes.

The majority of mapped footprints had a length of 29–31 nt, 30–32 nt, and 27–29 nt in human, mouse, and yeast, respectively. We assigned these mapped footprints to A-site codons according to the frame of the 5′ end of footprints[4]. For footprints with the 5′ end in the −1 frame the A-site was defined as position 17–19 and the ones with 5′ ends in the 0 frame as position 16–18.

For ribosome occupancy at the beginning of ORFs, we normalized the coverage of A-site footprints by the average ribosome occupancy of all codons in that gene:

$$D_{ij} = \frac{F_{ij}}{(\sum_{j=1}^{L_i} F_{ij})/L_i} \tag{1}$$

Where $F_{ij}$ and $D_{ij}$ are the ribosome footprints and density of position $j$ of gene $i$, respectively. $L_i$ is the length of genes in codons. The average of ribosome densities at the position $j$ is calculated by normalizing to the number of all well-expressed genes (>64 reads) with an ORF length of at least 200 codons:

$$A_j = \frac{\sum_{i=1}^{N} D_{ij}}{N} \tag{2}$$

Where $A_j$ is the average of ribosome densities and $N$ is the number of genes.

A-site codon occupancy plots[8] and wave plots[33] were generated as described. Differential gene-expression analysis was performed using the DESeq2 package[47]. Count matrix for long and short footprints was generated using a custom script that excluded the first 15 codons of a transcript. For altered transcripts, the *padj* threshold (*alpha*) was set to 0.05. To identify unaltered transcripts, the *althypothesis* function was set to "lessAbs". Correlation plots were generated using corrplot package[73].

**Reporting summary**. Further information on research design is available in the Nature Research Reporting Summary linked to this article.

## Data availability
The data that support this study are available from the corresponding author upon reasonable request. The sequencing data from the ribosome profiling experiments used in this study are available in the Gene Expression Omnibus database under accession code GSE136940.

## Code availability
The description of the algorithms applied are found in the methods section. Source codes for codon-occupancy calculations are available on github (https://github.com/LeidelLab/Codon_occupancy_cal).

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

## Acknowledgements

We thank Claudia Gräf and Fiona Alings for technical support and sequencing library generation, Namit Ranjan, Juanma Vaquerizas and all members of the Leidel lab for comments on the manuscript and critical discussions. *Schizosaccharomyces pombe* strain 972 (*h⁻*) and *Candida albicans* strain SN87 were kindly provided by Elena Hidalgo and Suzanne Nobel, respectively. This work was supported by the Max Planck Society and DFG [LE 3260/3-1] to S.A.L; P.S., B.S.N. and J.W. gratefully acknowledge fellowships from the Graduate School of the Cells-in-Motion Cluster of Excellence [EXC 1003-CiM]. Part of this work was a contribution to the BMBF funded project: MOSQUIVIR (FKZ 031L0127).

## Author contributions

P.S., J.W., B.S.N., and S.A.L. conceived the project and designed the experiments. P.S. and B.S.N. performed the ribosome profiling experiments and analyzed the data. J.W. performed computational analyses and wrote the source codes. All authors wrote the manuscript.

## Competing interests

The authors declare no competing interests.
