## [Peer Review File · Nature Communications]

Humans and other commonly used model organisms are resistant to cycloheximide-mediated biases in ribosome profiling experimentsReviewers' Comments:

Reviewer #1:

Remarks to the Author:

Ribosome footprint profiling provides genome-wide views of translation at the resolution of single codons and has revolutionized the field of translational regulation. Sharma et al provide a systematic analysis of the effects of a common translational inhibitor, cycloheximide (CHX) in the ribosome profiling protocol. The use of this inhibitor has been the subject of much debate in the ribosome profiling community, and has led to a number of erroneous conclusions, particularly in early work in the field. Their results suggest that use of this inhibitor may be of less concern in mammalian datasets compared to the yeast ones where it has been more thoroughly examined. The analyses and commentary in the manuscript also provide a starting point for standardization of this complex and highly varied technique. Moreover, the optimized digestion conditions that they present are a potential solution to the ongoing problem of high rRNA contamination in ribosome profiling libraries. This manuscript will be of interest to anyone performing ribosome profiling or analyzing the resulting datasets and should be published. However, I think that the manuscript could benefit from more careful discussion of the study limitations, as detailed below.

Issues and comments:

It is not clear what size of ribosome footprints was selected on the first size selection gel for ribosome profiling in various figures. This has important implications for the viability of using increased RNase I amounts as a method to deplete rRNA. In supplementary figure 1, where the extent of rRNA contamination is shown, only footprints of length 27-32 are shown, while lengths 19-32 are shown in figure 1. I am assuming that for S1, only 27-32nt footprints were isolated and that the 19-32nt libraries in F1 have much higher rRNA contamination.

How can CHX treatment dramatically affect E-site occupancy without affecting P or A site occupancy? Does the difference in E-site occupancy not imply that the ribosomes are in different positions upon CHX treatment?

On the lack of detection of ribosomal waves in species other than *S. cerevisiae*: The CGA codon in baker's yeast is not slowly translated due to its cognate tRNA's low abundance, but due to its use of I:A wobble pairing in decoding. This was shown by work in Elizabeth Grayhack's Lab, where the rate of decoding of CGA cannot be rescued by overexpressing the cognate tRNA for CGA codons but can be rescued by a mutant tRNA that utilizes Watson-Crick pairing. Thus, the statement on line 246 that "CGA and CGG are rare codons in these species and are decoded by only one tRNA each" may not be relevant. Do we know that CGA codons are slowly decoded in these other species even with only 1 genomic tRNA copy? Are the A-site occupancies for CGA comparable to those in *S. cerevisiae*? Without a set of well-defined slow translation events to examine, the conclusion that CHX is not causing waves downstream of slow codons (and in fact all codons) is not supported. It is possible that in rich-media conditions where there are not large differences between codons, there is little detectable effect of CHX pretreatment. However, CHX pretreatment could cause more serious artifacts in starvation or other conditions that strongly affect translation of specific codons. These points should be carefully considered in this section and in the discussion, as these results may lead some researchers to use CHX pretreatment in situations where it could affect experimental conclusions.

An aspect of CHX treatment that was not considered in this manuscript is the effect of CHX on ribosome accumulations on 5' UTR uORFs, which is artifactually increased by CHX pretreatment (Kearse et al, *Genes & development* 2019).

Figure 5 – Given the well-known genomic instability of cell lines such as HEK293, as well as the susceptibility of overall translation to often unreported factors such as passage number, media age, cell density, etc. it is perhaps unsurprising that HEK datasets are poorly correlated between labs compared to yeast datasets. The major differences could well be underlying biology rather than differences in library preparation. For example – is there a separation between HEK293 and HEK293T datasets? Or any other identifiable factor?

Could the overall difference between susceptibility of A site codon occupancy to CHX come down to differences in elongation rate or rate-limiting elongation step between yeast and mammals?

The work of Santos et al. (NAR 2019) may be worth commenting on. They see dramatic differences in TEs of some genes upon CHX pretreatment.

Minor issues/comments:

The phrase “nor does CHX affect global translation” in the abstract should be reworded. As is, it sounds like the authors are claiming that CHX does not affect translation at all in human cells, when we of course know it is a potent inhibitor of translation. I think that the authors mean that CHX doesn’t affect gene-level measures of ribosome occupancy, or something like that?

Figure 1 caption – the title says yeast libraries, but the figure only looks at HEK293T cells

Figure 3 – axis label says A-site codon occupancy on panels that do not look at the A site.

Figure 4 – the heatmap was uninterpretable blurry in one of my PDF viewing programs.

Reviewer #2:

Remarks to the Author:

The work by Sharma and colleagues systematically varies some technical parameters in ribosome footprint profiling library construction and assesses their impact on the observed distributions of ribosome protected footprints. The authors generate a consistent dataset that may be useful for the field in considering the effects of certain technical parameters in ribosome footprint profiling experiments (e.g. RNase treatment).

However, the value of the study is severely limited by the choice to focus on relatively uninformative longer footprints. The title and abstract are misleading because the data do not demonstrate an absence of cycloheximide mediated biases in short footprints (~21mers) that have recently been shown to be more informative of biologically relevant ribosome pauses. The current study focuses almost exclusively on longer footprints (~28mers)—a regrettable choice that the authors do not present any evidence to justify. Whether or not these longer footprints are affected by cycloheximide is beside the point if they do not accurately reflect meaningful differences in elongation rates in cells.

Major points:

1. The analysis of 28nt FPs misses the point if one is interested in codon-level ribosome dynamics, which is a main reason to perform ribosome profiling. In human (HeLa) cells starved for the amino acid glutamine, slowed ribosomes at Gln codons are barely detectable in 28nt FPs but are prominent in 21 nt FPs (Wu et al. (Green) 2019, Fig. S3C). It may be correct that CHX “does not affect quantification of codon-specific ribosome occupancy” of 28mers, but what evidence is there that 28mers report accurately on biologically relevant changes to translation elongation? None is provided in the current manuscript.
2. It is not clear what research question the authors believe “necessitates the preparation of small footprints”. The data from Wu et al. 2019 strongly suggest that any time one cares about the rate of ribosome movement through specific codons (including studying the effects of tRNA modifications on codon-level translation, a focus of research by the current authors), it will be necessary to examine small footprints. The problem of rRNA contamination is substantial, but cutting narrowly around less informative footprint sizes is not a good solution! The current manuscript would mislead the field on this critical point.
3. The methods provided are inadequate. This is surprising given that a main point of the study is that

technical variation causes significant differences in the data. There is no information about how the libraries were prepared after the isolation of monosomes from sucrose gradients. What size markers were used to select footprint RNA? Without this information, observations about the observed distributions of footprint lengths are meaningless.

4. The authors should repeat all of the codon-level analyses of Fig. 3 and Fig. 4 using the 21nt footprints and determine the correlations to tAI, rare codons, etc.

Additional points:

It is not clear what justifies the strongly worded conclusion that "Cycloheximide does not affect the translation ramp" (Fig. 2). The normalized coverage profiles in Fig. 2A are strikingly different dependent on treatment (+/+ , -/+ , -/- cycloheximide).

Fig. 2A shows a >2-fold accumulation of ribosomes at stop codons in the (-/+) sample compared to the others. This observation strongly suggests there is significant ribosome movement through the CDS in the absence of CHX added to the cells. It is not clear which of the treatments produces the best approximation of the desired 'snapshot' of ribosome locations in vivo.

**Reviewer #1 (Remarks to the Author):**

*Ribosome footprint profiling provides genome-wide views of translation at the resolution of*
*single codons and has revolutionized the field of translational regulation. Sharma et al provide*
*a systematic analysis of the effects of a common translational inhibitor, cycloheximide (CHX)*
*in the ribosome profiling protocol. The use of this inhibitor has been the subject of much debate*
*in the ribosome profiling community, and has led to a number of erroneous conclusions,*
*particularly in early work in the field. Their results suggest that use of this inhibitor may be of*
*less concern in mammalian datasets compared to the yeast ones where it has been more*
*thoroughly examined. The analyses and commentary in the manuscript also provide a starting*
*point for standardization of this complex and highly varied technique. Moreover, the optimized*
*digestion conditions that they present are a potential solution to the ongoing problem of high*
*rRNA contamination in ribosome profiling libraries. This manuscript will be of interest to*
*anyone performing ribosome profiling or analyzing the resulting datasets and should be*
*published. However, I think that the manuscript could benefit from more careful discussion of*
*the study limitations, as detailed below.*

We thank this reviewer for the positive assessment of our work and the suggestions to
improve our manuscript.

*Issues and comments:*

*1. It is not clear what size of ribosome footprints was selected on the first size selection gel for*
*ribosome profiling in various figures. This has important implications for the viability of using*
*increased RNase I amounts as a method to deplete rRNA. In supplementary figure 1, where the*
*extent of rRNA contamination is shown, only footprints of length 27-32 are shown, while*
*lengths 19-32 are shown in figure 1. I am assuming that for S1, only 27-32nt footprints were*
*isolated and that the 19-32nt libraries in F1 have much higher rRNA contamination.*

We agree that we have not sufficiently explained our strategy of footprint size selection
in material and methods and in the figures. We have amended this by updating material
and methods and the figure legends to clarify the size of the footprints used for our
analyses. Briefly, all figures in the manuscript that use our own data are based on libraries
prepared from footprints in the range of 18-32 nt (HEK 293T) and 18-30 nt
(*S. cerevisiae*). These footprints were cut from polyacrylamide gels using size specific
markers of 18, 30 and 32 nt.

The text in material and methods now reads:

All libraries were prepared from footprints in the range of 18-32 nt (HEK 293T) and
18-30 nt (*S. cerevisiae*). These footprints were cut from acrylamide gels using RNA
size markers of 18 nt and 30 nt (yeast) or 18 nt and 32 nt (HEK 293T).

We added the following sentence to the legend of Fig. 1:

Footprints were excised between 18-32 nt.

We added the following sentence to the legend of Fig. 2:

Uniquely mapped reads were split into short footprints (bottom left; 21-22 nt) and
long footprints (top right; 28-32 nt).

We added the following sentence to the legend of Fig. 3:
Footprints were excised between 18-32 nt but are represented according to size: short
footprints (21-22 nt) and long footprints (29-31 nt).

We added the following sentence to the legend of Fig. 4:
This plot only uses long footprints (29-31 nt).

We added the following sentences to the legend of Supplementary Fig. 1:
Published^{1,2} yeast libraries and libraries using higher RNase I concentration were
generated from long (28-30 nt) footprints using a similar protocol. [...] Footprints
were excised between 18-30 nt (yeast) and 18-32 nt (HEK 293T).

We added the following sentences to the legend of Supplementary Fig. 2:
63 A) Normalized ribosomal A-site coverage observed in short footprints (21-22 nt) for
the first 200 (left) and last 200 (right) codons in HEK 293T cells [...] B) Normalized
ribosomal A-site coverage for long footprints of HEK 293T cells. [...] C) Normalized
ribosomal A-site coverage for long footprints of human ORFs [...] D) Normalized
ribosomal A-site coverage for long footprints (27-29 nt) for the first 200 (left) and last
200 (right) codons in yeast cells in highly expressed genes (>64 reads).

We added the following sentences to the legend of Supplementary Fig. 4:
Transcriptome-wide ribosome enrichment profiles surrounding CGA and UUA
codons according to Hussmann *et al.*¹³ for short footprints (21 and 22 nt) of HEK 293T
cells using different CHX treatment regimens.

Supplementary Fig. 1A now shows the complete footprint length distribution.

Supplementary Fig. 1A: A) Histograms showing the effect of RNase I concentration
on footprint length and reading frame in human (-/-) ribosome profiling libraries (18-
32 nt footprints).

Supplementary Fig. 1C shows that rRNA levels decrease upon stronger digestion of
samples prepared from 18-32 nt footprints. Following the request of reviewer 2 we now
provide independent analyses of small (21-22 nt) and large (28-32 nt) footprints in our

figures. Therefore, we have expanded our contaminant analysis and now characterize
 additional classes of non-rRNA contaminants that are found in the size range of small
 footprints and long footprints. This information is included in Supplementary Fig. 1D.
 Note the fraction of miRNA that is only seen at the size of short footprints, which can
 serve as a positive control.

 **Supplementary Fig. 1D: Stacked barplots representing various non-rRNA**
 **contaminants in reads which did not map to rRNA and CDS under different RNase I**
 **concentrations.**

 *2. How can CHX treatment dramatically affect E-site occupancy without affecting P or A site*
 *occupancy? Does the difference in E-site occupancy not imply that the ribosomes are in*
 *different positions upon CHX treatment?*

 This is an interesting question. The E-site is generally not considered important for
 mRNA decoding or peptide bond formation. However, the E-site tRNA needs to be
 ejected to enable binding of the next tRNA in the A-site. Therefore, it is a theoretical
 possibility that the E-site might influence the activities of the P-site and of the A-site in
 a codon-specific manner. In human samples, we did not observe CHX mediated effects
 on the A-site or the P-site, emphasizing that any potential effect at the E-site does not
 alter codon occupancy at these sites. Nevertheless, we reported the finding such that
 researchers, who focus on E-site biology, are aware of this. When interested in codon
 occupancy at the E-site, researchers will need to carefully consider the use of published
 CHX datasets. Importantly, the effect only occurs following pre-incubation with CHX,
 which we do not recommend as a general strategy.

As for the potential cause of the effect, we hypothesize that the identity of the codon-
 anticodon pair that occupies the E-site alters the binding affinity of CHX for specific
 conformations of translating ribosomes. However, to clarify this point will require
 extensive structural work, which is of great interest for future studies but beyond the
 scope of this manuscript. We have performed the new analyses for the small ribosome
 footprints and have added their correlation of codon occupancy at the E/P/A-sites to
 Fig. 3. Similar to long footprints, small footprints show a slightly weaker correlation at
 the E-site between samples that were pre-incubated with CHX (+/+) and the samples that
 were not pre-treated (-/- and -/+). Finally, we extend our mention of E-site codon

occupancy in the results and specifically write in our discussion that analyzing E-site biology will require special caution.

Fig. 3: Cycloheximide (CHX) does not alter mammalian ribosome occupancy at the A-site and the P-site. A) Correlation analysis of transcriptome-wide A-site codon occupancy in HEK 293T cells across different CHX treatments for short and long footprints (mean; n=3). Each black dot represents a codon. The size of the box indicates p-values. Correlations with a p-value > 0.05 are crossed out. B) Like A) for P-site codon occupancy. C) Like A) for E-site codon occupancy. Footprints were excised between 18-32 nt but are represented according to size: short footprints (21-22 nt) and long footprints (29-31 nt).

The new text in the results reads:

To test for the CHX-dependent enrichment of codons, we calculated transcriptome-wide codon occupancy for the A-, P- and E-site codons in human and yeast (Fig. 3A-3C, Supplementary Fig. 3A and data not shown). For short and long footprints, correlations of A-site-codon occupancy in HEK 293T libraries were very high, independent of the treatment ($R^2 \geq 0.92$ between all conditions; Fig. 3A). Similarly, P-site-codon occupancy correlated well between treatments ($R^2 \geq 0.74$; Fig. 3B) while CHX-pre-treatment changed E-site-codon occupancy markedly ($R^2 = 0.28-0.35$ for

++ and -- conditions; Fig. 3C). A-site codon occupancy in human cells was not
affected by the digestion strength during footprint generation (Supplementary Fig.
3B). Similarly, the duration of CHX incubation or the use of a different cell line (HeLa
instead of HEK 293T) only played a minor role (Supplementary Fig. 3C). This shows
that codon-specific analyses except for the E-site can be conducted in human cells
independent of CHX treatment.

The new text in the discussion reads:

There are three conditions that require specific attention when using cycloheximide:
First, E-site codon occupancy is affected by pre-treatment with CHX in human cells.
Generally, the E-site is not considered critical for decoding or peptide-bond formation.
However, for researchers interested in E-site biology, the use of CHX is not
recommended. Since CHX binds to the E-site its presence likely alters the E-site
interaction with tRNA thus affecting tRNA release.

3. On the lack of detection of ribosomal waves in species other than *S. cerevisiae*: The CGA
codon in baker's yeast is not slowly translated due to its cognate tRNA's low abundance, but
due to its use of I:A wobble pairing in decoding. This was shown by work in Elizabeth
Grayhack's Lab, where the rate of decoding of CGA cannot be rescued by overexpressing the
cognate tRNA for CGA codons but can be rescued by a mutant tRNA that utilizes Watson-Crick
pairing. Thus, the statement on line 246 that "CGA and CGG are rare codons in these species
and are decoded by only one tRNA each" may not be relevant.

We thank the reviewer to ask us to clarify this point. Indeed, the groundbreaking work
of Elizabeth Grayhack's laboratory has established that the suboptimal properties of CGA
codons during translation mainly stem from I34•A3 wobble interaction between tRNA^{Arg}_{ICG}
and CGA (Letzring *et al.*, 2010). Overexpression of tRNA^{Arg}_{ICG} rescues the effect only to a
small degree while an artificial tRNA^{Arg}_{UCG} that does not exist in yeast rescues completely
(Letzring *et al.*, 2010).

Following our unexpected finding that CGA does not cause a 'ribosomal wave' in humans
and other species we asked ourselves, whether the abundance of tRNA^{Arg}_{ICG} and tRNA^{Arg}_{UCG}
plays a role in species related to *S. cerevisiae*. Hence, we analyzed *S. pombe* and
*C. albicans*. Both species do not show a ribosomal wave (Fig. 4). *S. pombe* contains a
single copy of tRNA^{Arg}_{UCG} able to decode CGA by Watson-Crick pairing. However,
*C. albicans* does not contain tRNA^{Arg}_{UCG} but requires tRNA^{Arg}_{ICG} to decode CGA like the
closely related *S. cerevisiae* (Chan and Lowe 2016). Furthermore, *C. albicans* contains
only two copies of tRNA^{Arg}_{ICG} (*S. cerevisiae* contains 6 copies), which according to
Letzring *et al.* is expected to further decrease the translation efficiency of CGA (Letzring
*et al.*, 2010). Interestingly, Letzring and coworkers have identified Rpl1b as a suppressor
of CGA defective translation. However, they did not test overexpression of tRNA^{Arg}_{ICG} in
this context (Letzring *et al.*, 2013). These findings suggest that a structural component
mediates CGA decoding and that CHX binding to the ribosome may be influenced by
the codon-anticodon interactions in the P- and A-site.

We have rewritten the paragraph in the results and in the discussion, to better explain the
CGA-codon effects. In this context we reference the work by the Grayhack laboratory by
additional references.

The text in the results now reads:

These ribosomal 'waves' occur downstream of the rare CGA and CGG codons. CGG
is decoded by the essential tRNA^{Arg}_{CCG}, which is expressed from a single gene in baker's
yeast^{54,55}. CGA exhibits a more pronounced wave than CGG and is read by tRNA^{Arg}_{ICG}
requiring a wobble interaction to be decoded^{56,33}. As expected, we observed a wave
for CGA and CGG in the +/+ but not in the -/+ and -/- libraries in yeast (Fig. 4A, and
data not shown). However, we did not observe a similar effect for CGA in HEK 293T
cells (Fig. 4B and Supplementary Fig. 4). In humans, decoding of CGA relies on its
cognate tRNA^{Arg}_{UCG} and does not require wobble interactions^{54,55}. Furthermore, there
are no codons in humans that are decoded by a single tRNA like tRNA^{Arg}_{CCG} in yeast
(Supplementary Table 1). Consistently, we did not observe wave formation for any
codon in humans (data not shown). Even for the UUA codon that is both rare and has
a low tRNA copy number in humans, we did not detect CHX-induced downstream
enrichment of ribosomes (Fig. 4B and Supplementary Fig. 4). These ribosomal waves
are not only absent from our human-cell libraries, but from all published human
ribosome profiling datasets that we analyzed (Fig. 4C). Interestingly, this effect is not
specific to human cells, because ribosomal waves are absent from ribosome profiling
data of other vertebrates (Fig. 4A). Neither zebrafish¹⁰ nor mouse⁵ show ribosome
enrichment downstream of any codon. Finally, to investigate whether altered
translation at rare codons is a general feature of yeasts we generated ribosome
profiling libraries from CHX treated *Schizosaccharomyces pombe* and *Candida*
*albicans*. Like in *S. cerevisiae*, CGG is decoded by a single copy of tRNA^{Arg}_{CCG} in both
species. However, while CGA is decoded by its cognate tRNA^{Arg}_{UCG} in *S. pombe*, it
depends on decoding by tRNA^{Arg}_{ICG} in *C. albicans*. Surprisingly, we did not observe
ribosomal waves at CGA, CGG or any other codon in both yeasts (Fig. 4D and data
not shown). This shows that ribosomal waves are specific to baker's yeast and that the
use of CHX is, therefore, mainly a concern when studying translation in *S. cerevisiae*,
but not in other widely used eukaryotic model organisms.

The text in the discussion now reads:

A codon-specific alteration of ribosome occupancy leading to ribosomal waves is
observed for CGA and CGG in CHX-pre-treated baker's yeast³³. These codons, which
are rare in *S. cerevisiae*, *S. pombe* and *C. albicans*, exhibit either low levels of cognate
tRNA or depend on wobble decoding⁵⁶. Hence, their tRNA adaptation index (tAI)
values, a measure of codon-anticodon decoding that takes into account tRNA copy
numbers and wobble-pairing constraints⁶⁴, are very low. In vertebrates, CGA is not a
rare codon and its tAI values and elongation rate are intermediate (data not shown).
Furthermore, in common vertebrate models, no codon features a similarly low codon
frequency and depends on wobble decoding like CGA in yeast, suggesting that these
organisms are less likely to be affected by CHX treatment. Consistently, we found no
evidence of ribosome enrichment downstream of any codon in zebrafish, mice and
humans. It would require more replicates and deeper sequencing to detect subtle
effects in specific ORFs or for specific codon pairs. In baker's yeast the inefficient
decoding of CGA largely depends on wobble pairing to tRNA^{Arg}_{ICG} with a minor
contribution of tRNA abundance⁵⁶. However, neither low codon frequency, low tRNA
copy numbers nor wobble codon-anticodon interactions can explain CHX-mediated
waves as single factors as shown by the comparison with *S. pombe* and *C. albicans*
(Fig. 4D). It cannot be excluded that differences in elongation rate or a rate-limiting

elongation step between yeast and mammals lead to the observed differences.
However, the comparison to *S. pombe* and *C. albicans* makes this less likely. Baker's
yeast may vary from other species by its uptake and turnover of CHX or by subtle
structural features of the yeast ribosome and how CHX interacts with it. We excluded
slower uptake of CHX in human cells as a contributing factor by increasing CHX
concentrations and incubation time (Fig. 1C). A recent study reported that the binding
site of CHX in human ribosomes is similar to yeast, but that the ligand adopts a
different conformation⁶⁵. This might facilitate a more stable binding of CHX to human
ribosomes leading to a better preservation of their position on mRNA. Interestingly, a
*rpl1bD* mutant was shown to improve translation of CGA codons pointing towards
structural constraints of yeast ribosomes⁶⁶. Therefore, special care will be required
when analyzing CGA. However, this codon also triggers phenotypes in orthogonal
assays^{56,67,68}. Hence, the observed effects likely indicate a true biological feature. To
disentangle the molecular mechanisms will necessitate a detailed analysis of CHX-
binding to ribosomes in more organisms.

4. Do we know that CGA codons are slowly decoded in these other species even with only one
genomic tRNA copy? Are the A-site occupancies for CGA comparable to those in *S. cerevisiae*?
Without a set of well-defined slow translation events to examine, the conclusion that CHX is
not causing waves downstream of slow codons (and in fact all codons) is not supported.

tRNA copy numbers are highly correlated with tRNA abundance both in bacteria and
eukaryotes (dos Reis *et al.*, 2004). Furthermore, the translation speed of codons decoded
by abundant tRNAs is fast and transcripts enriched for these codons yield more protein
(Tuller *et al.*, 2010). The tAI values of CGA codons, which are computed based on tRNA
copy numbers, are very small in *S. pombe*, *C. albicans* and in *S. cerevisiae*. However,
when ranking all codons according to A-site occupancy, CGA is not amongst the top-
ranked codons in terms of occupancy in *S. pombe* and *C. albicans*. This is conceivable,
since translation speed is not only determined by tRNA abundance but also by the coding
sequence including codon position, mRNA secondary structure etc. A high A-site
occupancy of CGA is only found in *S. cerevisiae* both in the -/+ or -/- samples.
Furthermore, Hussmann *et al.* (Hussmann *et al.*, 2015) analyzed the data of a tRNA
modification mutant from Zinshteyn and Gilbert (Zinshteyn and Gilbert, 2014) and found
a ribosomal wave downstream of the slow codon AAA. Based on this observation, we
hypothesized that the translation speed is a key factor for causing ribosomal waves.
However, we did not observe a ribosomal wave for CCG, which appears similarly slow
as CGA in our -/+ samples. Thus, we assume that codon speed is not be the only trigger
for ribosomal waves. Likely structural features contribute to these effects (see our reply
above).

Finally, in humans and mice, CGA is not a very rare codon and its tAI values and
elongation rate are intermediate. We have searched for potential waves downstream of
every codon in the species that we have analyzed. However, we did not find evidence for
a wave similar to CGA in baker's yeast downstream of any codon. Since we used the
method described by Hussmann *et al.* (Hussmann *et al.*, 2015), we expect that our
analysis would have identified such effects if they existed. We did not look at ribosomal
waves following individual codons on specific transcripts. However, ribosomal waves
are a codon-specific phenomenon that becomes apparent in metagene analyses. To
observe effects like this in a single transcript would require an extremely deep
sequencing, if at all possible.

We discuss this point in the manuscript:

A codon-specific alteration of ribosome occupancy leading to ribosomal waves is
observed for CGA and CGG in CHX-pre-treated baker's yeast³³. These codons, which
are rare in *S. cerevisiae*, *S. pombe* and *C. albicans*, exhibit either low levels of cognate
tRNA or depend on wobble decoding⁵⁶. Hence, their tRNA adaptation index (tAI)
values, a measure of codon-anticodon decoding that takes into account tRNA copy
numbers and wobble-pairing constraints⁶⁴, are very low. In vertebrates, CGA is not a
rare codon and its tAI values and elongation rate are intermediate (data not shown).
Furthermore, in common vertebrate models, no codon features a similarly low codon
frequency and depends on wobble decoding like CGA in yeast, suggesting that these
organisms are less likely to be affected by CHX treatment. Consistently, we found no
evidence of ribosome enrichment downstream of any codon in zebrafish, mice and
humans. It would require more replicates and deeper sequencing to detect subtle
effects in specific ORFs or for specific codon pairs.

*5. It is possible that in rich-media conditions where there are not large differences between*
*codons, there is little detectable effect of CHX pretreatment. However, CHX pretreatment could*
*cause more serious artifacts in starvation or other conditions that strongly affect translation*
*of specific codons. These points should be carefully considered in this section and in the*
*discussion, as these results may lead some researchers to use CHX pretreatment in situations*
*where it could affect experimental conclusions.*

We thank this reviewer for this comment. Indeed, we cannot exclude that CHX-
pretreatment could cause differences in codon occupancy under specific conditions and
in certain species. In contrast to human cells, we saw an influence of CHX-pre-treatment
in baker's yeast. These effects concern ribosome occupancy around the start codon, at the
E-site and during termination. Growth conditions that affect these processes are likely
influenced by CHX-pretreatment. However, codon occupancy is a feature that measures
translation elongation and not initiation. Currently, we do not have evidence that the use
of CHX in the buffer (-/+) would affect measurements in a similar way.

If translation elongation *per se* is altered under specific conditions this will have to be
mediated e.g., through changes in tRNA modifications, tRNA charging, rRNA
modifications, mRNA modifications, the ribosome or translation factors. As far as we
know, such changes will likely result in an environmental stress response or trigger the
Gcn4 pathway and will be observable in the gene expression pattern. Hence, we advise
every researcher to specifically look for signs of stress in their data and to confirm that
the observed signal is not altered by CHX by conducting -/- experiments. However, when
performing experiments without CHX all other experimental conditions like e.g., the
harvesting method need to be identical.

We discuss this point in our manuscript.

We performed our experiments under rich-media conditions in cells that were not
exposed to stress during culture and harvesting. We cannot exclude that CHX-
mediated effects occur under specific stress and in certain species. The step that
appears most vulnerable is initiation (Fig. 2A and Supplementary Fig. 2A and 2D).
However, stresses that affect translation initiation will very likely coincide with a
general stress response that will be visible in the gene expression pattern (e.g., by
induction of *GCN4* in yeast)^{39,69}. In particular in light of reports that found changes in
translation efficiency in response to starvation upon long pre-treatment with CHX, it

is advisable to verify findings by omitting CHX from experiments³⁶. However, at this
point, there is no evidence of negative effects of using CHX in the buffer (-/+).

6. An aspect of CHX treatment that was not considered in this manuscript is the effect of CHX
on ribosome accumulations on 5' UTR uORFs, which is artifactually increased by CHX
pretreatment (Kearse *et al.*, *Genes & Development* 2019).

We did not analyze the effect of CHX on the translation of uORFs in our manuscript
since we focused on translation elongation in the CDS. Most uORFs are 10-30 codons in
length and use non-canonical start codons (Lawless *et al.*, 2009). Hence, they were not
part of our analyses. Furthermore, CHX at concentrations used in most studies blocks
translation elongation but not translation initiation. Therefore, the pre-treatment with
CHX will lead to an accumulation of ribosomes at the initiation site or at uORFs. We had
noticed that this effect can be extensive when incubation times of >1 min are used
(Kearse *et al.* use 15 min or 24 h pre-incubation). Therefore, we recommend not to
exceed 1 min pre-incubation if CHX-pretreatment is required. However, in general we
do not recommend CHX pre-incubation but to only include it into the buffer (-/+).

We discuss this more explicitly in our manuscript, to avoid the false assumption that a
prolonged incubation with CHX does not affect the results. Like every component in an
experiment, also CHX needs to be used in a reasonable manner.

We added the following sentences to the discussion:

Therefore, we generally recommend the use of the -/+ protocol for human cells and
other species including yeast. Adding CHX to the lysis buffer provides the advantage
of limiting ribosomal run-off. If the experimental design requires CHX pre-
incubation, the exposure of the cells should be kept as brief as possible (≤ 1 min) to
avoid an accumulation of reads around the start unless this is intended.

Finally, we have now analyzed the reads mapping to the 5' UTR under the three treatment
conditions. We have used this strategy because if uORFs are generally affected by CHX
the deregulation of reads in 5' UTR should be easily observed and can be used as a proxy
for uORF translation.

We have added the data to the manuscript as Supplementary Fig. 2F and discuss it in the
results.

Next, we analyzed footprint density around annotated start and stop codons. The
enrichment of reads at the initiation site in response to CHX pre-treatment can be
exploited to identify upstream open reading frames (uORFs) or other features that are
associated with initiation. Recently, it was reported that translation initiation at non-
AUG start codons encoding for uORFs or N-terminal protein extensions is resistant
to CHX treatment³⁸. Therefore, we tested whether the different treatment conditions
influence ribosome occupancy in the 5' untranslated region (UTR) and compared the
reads mapping to the region upstream of annotated AUG start codons. Using long
footprints, we found 317 upregulated and 257 downregulated 5' UTRs in the +/+
condition compared to -/+. In contrast, comparing -/- and -/+ libraries revealed only 7
downregulated UTRs. For short footprints the low number of UTR reads did not allow
for a meaningful comparison. Thus, CHX pre-treatment alters the footprint density in
the 5' UTR and might be an interesting strategy to complement drugs like
harringtonine or lactimidomycin for uORF identification^{5,20}.

Supplementary Fig. 2F: Differential ribosome occupancy of 5' UTRs in HEK 293T cells across inhibitor treatments determined with DESeq2⁵. 5' UTRs were tested for differential translation (top right; adjusted p-value ≤ 0.05). Significantly altered 5' UTRs are indicated in blue.

7. Figure 5 – Given the well-known genomic instability of cell lines such as HEK293, as well as the susceptibility of overall translation to often unreported factors such as passage number, media age, cell density, etc. it is perhaps unsurprising that HEK datasets are poorly correlated between labs compared to yeast datasets. The major differences could well be underlying biology rather than differences in library preparation. For example – is there a separation between HEK293 and HEK293T datasets? Or any other identifiable factor?

We agree with this reviewer that it is very important to identify the source of variability in ribosome profiling data. With our analysis, we have excluded that CHX is a driver of variability in human cells. We have spent a considerable amount of time annotating various parameters in the published datasets that we analyzed to identify factors that can explain the observed differences. However, we did not identify such a factor. We now list several parameters that differ in the protocols that were used by the different studies analyzed in our manuscript as Supplementary Table 2. As rightfully pointed out by this reviewer, factors such as passage number of the cells, type of calf serum used or age of the cultivation media are generally not reported. Therefore, it was impossible to assess the impact of these factors on our correlation analysis. We consider it unlikely that a single biological factor is the driver of the observed differences between ribosome profiling libraries, because the observed deviations in quantifiable parameters of these datasets like ribosome footprint size, rRNA contaminations etc. are more easily explained by handling differences during library preparation.

Nevertheless, we have added this point to our results.

[revised manuscript text omitted]

Supplementary Fig. 3: Effects of different treatments on A-site codon occupancy.

A) Spearman correlations of A-site ribosome occupancy in yeast cells across different CHX treatments (mean, n=3). Each black dot represents a codon. Size of the box indicates p-value. Correlations with a p-value > 0.05 are crossed out. B) Same as A) for -/- HEK 293T cells treated with different RNase I concentrations. C) Same as A) for HEK 293T cells from this study and HeLa cells³. CHX-treatment conditions are indicated by color: +/+, green; +/-, yellow; -/-, pink.

Fig. 5: Cycloheximide (CHX) does not explain the poor correlation between various datasets in humans. A) Correlation analysis of transcriptome-wide A-site codon occupancy across data from this study and published datasets for human cells (HEK 293 and HEK 293T cells) using different CHX treatments^{19,20,58-63}. Each black dot represents a codon. The size of the box indicates p-value. Correlations with a p-value > 0.05 are crossed out.

Finally, we sought to estimate whether the HEK293 and HEK293T cells included in our analysis for Fig. 5 had genetically diverged between the different laboratories. Since we were unable to obtain genomic sequencing data, we attempted to estimate the genetic diversity by analyzing single nucleotide polymorphisms (SNPs) in these cells. However, the number of SNPs correlated mostly with sequencing library size and not with the origin from different laboratories. Thus, we concluded that diverging inter-laboratory evolution between these cell lines is unlikely to explain the differences in codon correlation, which is in line with our analysis of the HeLa cells.

We mention this point in the results:

We excluded that either the strength of digestion, the cell line or CHX incubation time are key factors that can individually explain the observed differences in A-site codon

occupancy (Supplementary Fig. 3B and 3C). Furthermore, we analyzed the genetic
diversity of the cells based on differences in single-nucleotide polymorphisms (SNPs).
However, we did not find evidence that the cells differed significantly (data not
shown). Therefore, our analysis suggests that a combination of factors causes these
differences and that the influence of these steps in the protocol is more relevant than
generally thought.

Importantly, even if such a single factor during cell culture or library generation existed
that could explain the observed differences, this is in line with our argument that CHX
mediated effects are not a major factor when analyzing ribosome profiling data in human
cells and likely in most other species.

8. *Could the overall difference between susceptibility of A site codon occupancy to CHX come*
*down to differences in elongation rate or rate-limiting elongation step between yeast and*
*mammals?*

We cannot exclude that the codon-specific difference in response to CHX between yeast
and mammals are due to differences in elongation rate or a rate-limiting elongation step
between these species. These are difficult to compare since ribosome profiling
measurements do not measure absolute values. However, we find this less likely, since
we did not observe ribosomal waves when analyzing other yeasts like *S. pombe* or
*C. albicans*.

Nevertheless, we mention this in our discussion:

*It cannot be excluded that differences in elongation rate or a rate-limiting elongation*
*step between yeast and mammals lead to this difference. However, the comparison to*
*S. pombe and C. albicans makes this less likely. Baker's yeast may vary from other*
*species by its uptake and turnover of CHX or by structural details of the yeast*
*ribosome and how CHX interacts with it.*

9. *The work of Santos et al. (NAR 2019) may be worth commenting on. They see dramatic*
*differences in TEs of some genes upon CHX pretreatment.*

The data of Santos *et al.* is not straight forward to interpret in this context. They use a
2 min CHX-pretreatment in baker's yeast. Furthermore, when comparing CHX vs. no
CHX the cultures were harvested differently. When pre-treating yeast with CHX the cells
were centrifuged. When CHX was omitted, cultures were harvested by rapid filtration.
This difference in harvesting likely affects the results independent of CHX. The
laboratory of Robbie Loewith (University of Geneva, Switzerland) has shown that even
brief centrifugation in otherwise rich-media conditions triggers TORC1-mediated
starvation response (Urban *et al.*, 2007).

The effect of CHX on the translation efficiency of ribosome biogenesis genes described
by Santos and coworkers is only observed in the samples under starvation. We reanalyzed
their data and found that, in their replete samples, CHX does not show an effect
(Reviewer Fig. 1, top right). Furthermore, we compared the translation efficiency
between *+/+* and *-/+* using the data from our lab and did not observe a CHX-dependent
effect on ribosome biogenesis genes (Reviewer Fig. 1, bottom). This is expected since
our samples are not under starvation. Finally, we have compared *GCN4* coverage in our
libraries and the Santos data. We did not notice differences in the induction of Gcn4 by
CHX pre-treatment (Reviewer Fig. 1, all panels).

Hence, we can confirm that prolonged pre-treatment with CHX can distort ribosome
translation measurements in yeast. Therefore, we generally do not recommend a pre-
treatment with CHX.

Reviewer Fig.: 1: Translation efficiency fold change vs mRNA fold change between
+/+ and +/- from our lab (Nedialkova and Leidel, 2015 and unpublished) and (Santos
et al.)(Ribo: Ribosome protein genes; Ribo_biogenesis, Ribosome biogenesis genes).

We briefly discuss the work of Santos *et al.* in the revised manuscript (see also point 5
above):

We performed our experiments under rich-media conditions in cells that were not
exposed to stress during culture and harvesting. We cannot exclude that CHX-
mediated effects occur under specific stress and in certain species. The step that
appears most vulnerable is initiation (Fig. 2A and Supplementary Fig. 2A and 2D).
However, stresses that affect translation initiation will very likely coincide with a
general stress response that will be visible in the gene expression pattern (e.g., by
induction of *GCN4* in yeast)^{39,69}. In particular in light of reports that found changes in
translation efficiency in response to starvation upon long pre-treatment with CHX, it
is advisable to verify findings by omitting CHX from experiments³⁶. However, at this
point, there is no evidence of negative effects of using CHX in the buffer (-/+).

*Minor issues/comments:*

10. The phrase “nor does CHX affect global translation” in the abstract should be reworded.
As is, it sounds like the authors are claiming that CHX does not affect translation at all in

*human cells, when we of course know it is a potent inhibitor of translation. I think that the*
*authors mean that CHX doesn't affect gene-level measures of ribosome occupancy, or*
*something like that?*

This is an important comment! We agree that our phrase can cause misunderstandings
and have therefore rephrased our abstract and other parts of the manuscript to avoid the
impression that CHX does not affect translation.

Our abstract now reads:

We find that human ribosomes are not susceptible to conformational restrictions by
CHX, nor does it distort gene-level measurements of ribosome occupancy, measured
decoding speed or the translational ramp.

11. *Figure 1 caption – the title says yeast libraries, but the figure only looks at HEK293T cells*

Thank you for catching this! We have corrected the mistake in the title, which stems from
an older version of the manuscript.

The new title is:

Cycloheximide (CHX) does not affect ribosome footprint length distribution in
HEK 293T cells.

12. *Figure 3 – axis label says A-site codon occupancy on panels that do not look at the A site.*

The axis label in Fig. 3 have been corrected. They read now:

Codon occupancy

13. *Figure 4 – the heatmap was uninterpretable blurry in one of my PDF viewing programs.*

We thank the reviewer for this hint! The heatmap was indeed blurry when using programs
like Preview on macOS. We have rebuilt Fig. 4 using a different file format, which has
fixed this issue in our hands.

**Reviewer #2 (Remarks to the Author):**

*The work by Sharma and colleagues systematically varies some technical parameters in*
*ribosome footprint profiling library construction and assesses their impact on the observed*
*distributions of ribosome protected footprints. The authors generate a consistent dataset that*
*may be useful for the field in considering the effects of certain technical parameters in ribosome*
*footprint profiling experiments (e.g. RNase treatment).*

*However, the value of the study is severely limited by the choice to focus on relatively*
*uninformative longer footprints. The title and abstract are misleading because the data do not*
*demonstrate an absence of cycloheximide mediated biases in short footprints (~21mers) that*
*have recently been shown to be more informative of biologically relevant ribosome pauses. The*
*current study focuses almost exclusively on longer footprints (~28mers)—a regrettable choice*
*that the authors do not present any evidence to justify. Whether or not these longer footprints*
*are affected by cycloheximide is beside the point if they do not accurately reflect meaningful*
*differences in elongation rates in cells.*

We thank this reviewer for the constructive criticism and suggestions that helped to
improve our manuscript.

*Major points:*

*1. The analysis of 28nt FPs misses the point if one is interested in codon-level ribosome*
*dynamics, which is a main reason to perform ribosome profiling. In human (HeLa) cells*
*starved for the amino acid glutamine, slowed ribosomes at Gln codons are barely detectable*
*in 28nt FPs but are prominent in 21 nt FPs (Wu et al. (Green) 2019, Fig. S3C). It may be*
*correct that CHX “does not affect quantification of codon-specific ribosome occupancy” of*
*28mers, but what evidence is there that 28mers report accurately on biologically relevant*
*changes to translation elongation? None is provided in the current manuscript.*

We thank this reviewer for raising this important point. The analysis of small footprints
has been pioneered by Lareau *et al.*, 2014 and Wu *et al.*, 2019 to capture specific
conformational states during the ribosomal elongation cycle and has provided important
insights into the translation process.

We have extended our analysis to short footprints whenever possible. They are now
included in Fig. 2B, 3A-3C and Supplementary Fig. 1D, 1O, 2A, 4, 5C and 5D. We were
unable to analyze short footprints in yeast experiments, since we only observed
meaningful amounts of short reads in the -/- sample. This made direct comparisons
impossible. Similarly, we did not include short footprints in the analysis of reads in the
5' UTR in human cells due to low read numbers. Finally, we could not include short
footprints in our analyses of published vertebrate datasets (Fig. 4 and Fig. 5) as these
studies capture only ~30nt long footprints.

In general, the results for short footprints were similar to those of long footprints. To
show that the analysis of long footprints according to the classical protocol is justified,
we reanalyzed the data of Wu *et al.* for long and short footprints (Supplementary Fig.
1O). While the Gln-specific signal is stronger for short footprints in humans, it is less
noisy for long footprints such that the result is similarly clear. The difference appears to
be of more quantitative than qualitative nature.

Supplementary Fig. 10: Codon specific changes in A-site ribosome occupancy for short and long footprints in glutamine starved HeLa cells lysed with buffer containing either 100 $\mu\text{g/ml}$ CHX or cocktail of 100 $\mu\text{g/ml}$ CHX and 100 $\mu\text{g/ml}$ TIG⁴.

We mention this in the results:

During the translation elongation cycle ribosomes adopt several conformations. Nevertheless, only two distinct footprint sizes can be isolated in ribosome profiling experiments^{32,37}. Recently, cocktails of translation inhibitors were introduced to enrich for specific steps of translation elongation³⁷. In yeast libraries that used inhibitor cocktails short footprints appear to more accurately reflect translation than classical long footprints, while this difference is less striking if only CHX is used³⁷. To assess the situation in human cells, we analyzed short and long footprints of the published HeLa datasets³⁷. Interestingly, HeLa cells starved for glutamine show stronger codon-specific effects in short footprints due to perturbed charging of $\text{tRNA}_{\text{UUG}}^{\text{Gln}}$ and $\text{tRNA}_{\text{CUG}}^{\text{Gln}}$ (Supplementary Fig. 10, left). However, the long footprint

data showed the same result. While the scale of the effect was weaker, the data were
less noisy for long footprints (Supplementary Fig. 1O, right). Since the
implementation of ribosome profiling, hundreds of studies have analyzed long
footprints. Furthermore, analyses of codon-specific perturbations by using tRNA
modification mutants have shown that long footprints provide robust information
about translational slowdown^{8,39-43}. Therefore, we conducted our analyses for long
footprints and included short footprints for comparison whenever possible.

And in the discussion:

Recently, short footprints have entered the stage in addition to the canonical long
footprints as a tool to obtain more information about ribosomal conformations^{32,37}. In
yeast, short footprints show a high anti-correlation with the tAI. However, no
significant correlation with 1/tAI is seen in human cells (Supplementary Fig. 5C).
Similarly, the correlation between codon occupancy and amino acid polarity observed
for short footprints in yeast³² is absent in human cells (Supplementary Fig. 5D;
Hydrophobicity index). In fact, we did not find significant correlation between codon
occupancy and most of the biochemical properties of amino acids for both long and
short footprints (Supplementary Fig. 5D). Short footprints either derive from rotated
ribosomes or from unrotated pre-accommodation-state ribosomes with an empty A-
site^{32,37}. In libraries that are devoid of CHX short and long footprints simultaneously
occur in yeast³², while both types of footprints are present in human libraries
independent of the CHX regimen (Fig. 1B and Supplementary Fig. 1H-1M). While
empty A-sites can be induced by treatments like specific amino-acid starvation or the
use of tRNA toxins³⁷, it is difficult to distinguish these two types of short footprints
*in vivo*. By combining CHX with other inhibitors like tigecycline or anisomycin in
yeast specific types of short footprints can be enriched, allowing to further improve
the resolution of ribosome profiling³⁷. Nevertheless, the use of canonical long
footprints and CHX have allowed to identify biological meaningful features in
ribosome profiling experiments e.g. in the analysis of tRNA modification mutants8,39-
41,43. We have analyzed several features in ribosome profiling data in response to CHX
exposure independently for small and large footprints and found the results to be very
similar. While we cannot exclude that subtle differences exist, these appear to be
rather of quantitative than of qualitative nature.

*2. It is not clear what research question the authors believe “necessitates the preparation of*
*small footprints”. The data from Wu et al. 2019 strongly suggest that any time one cares about*
*the rate of ribosome movement through specific codons (including studying the effects of tRNA*
*modifications on codon-level translation, a focus of research by the current authors), it will be*
*necessary to examine small footprints. The problem of rRNA contamination is substantial, but*
*cutting narrowly around less informative footprint sizes is not a good solution! The current*
*manuscript would mislead the field on this critical point.*

In our opinion the preparation of small footprints is recommended for experiments, in
which a high frequency of empty A-sites is expected or when using inhibitor cocktails in
yeast.

Wu *et al.* show that short footprints are an interesting tool to yield additional information
in ribosome profiling experiments. However, there are two points that require caution
when extending conclusions from the Wu *et al.* data to other situations. First, the biggest
advantage of using short footprints occurs in yeast when using cocktails of inhibitors,
while the difference between short and long footprints is less striking when using CHX

only (Wu *et al.*, 2019; Fig. 2C). Second, all three experimental paradigms used by Wu *et al.*
 that support the physiological relevance of short footprints lead to an increase in
 frequency of empty A-sites:

First, in yeast Wu *et al.* used the γ -toxin to cleave tRNA_{UUC}^{Glu}, which decodes GAA
 codons. Consequently, they observed an increase in the proportion of 21 nt footprints
 with GAA in the A-site. An increase is also seen in 28 nt footprints albeit to a lesser
 extent (Reviewer Fig. 2; Bottom panels). The γ -toxin is a very powerful tool to disrupt
 codon-specific translation but does not reflect a physiological situation. By cleaving
 tRNA_{UUC}^{Glu}, the γ -toxin dramatically alters the cellular tRNA pool. Hence, the availability
 of this tRNA is significantly reduced leading to an increase of ribosomes with a GAA
 codon in the empty A-site. Therefore, a strong signal of GAA in 21 nt footprint is
 expected.

Second, similarly in yeast, the authors used 3-amino-1,2,4-triazole (3-AT) a competitive
 inhibitor of histidine synthesis. High doses of 3-AT block histidine synthesis. Therefore,
 aminoacyl-tRNA synthetases fail to charge tRNA_{GUG}^{His}, which decodes the histidine
 codons CAC and CAU. This leads to an increased frequency of ribosomes with empty
 A-sites at CAC and CAU codons. Thus, an increase of short footprints with these codons
 in the A-site is expected (Reviewer Fig. 2; Top panels).

 **Reviewer Fig. 2: Correlation analysis of transcriptome-wide A-site codon occupancy**
 **for Wu *et al.*, 2019 across different treatments for short and long footprints (top: 3-**
 **AT treatment; bottom: tRNA cleavage using the γ -toxin).**

 Third, to induce a codon-specific translation defect in human cells, Wu *et al.* starve HeLa
 cells for several hours for glutamine. This leads to uncharged tRNA_{UUG}^{Gln} and tRNA_{CUG}^{Gln}
 and subsequently to empty A-sites like in the 3-AT paradigm. Thus, this strategy
 similarly enriches for its signal in 21 nt footprints due to the lack of charged tRNAs.
 However, in our reanalysis of their data the slowdown at glutamine codons is also

distinctly observable in 28 nt footprints (Supplementary Fig. 10). While the signal is stronger for short footprints, it is less noisy in long footprints.

Supplementary Fig. 10: Codon specific changes in A-site ribosome occupancy for short and long footprints in glutamine starved HeLa cells lysed with buffer containing either 100 $\mu\text{g/ml}$ CHX or cocktail of 100 $\mu\text{g/ml}$ CHX and 100 $\mu\text{g/ml}$ TIG⁴.

Importantly, while these three conditions are expected to induce empty A-sites and therefore short footprints, this does not imply that short footprints are more relevant than 28 nt footprints under physiological conditions. Importantly, as shown in our new Fig. 2B and 3, Supplementary Fig. 2A and 4, there is no major difference between short and long footprints, suggesting that this concern is unnecessary.

In the past, several laboratories including our own have successfully used ribosome profiling to analyze codon-specific translation rates of non-essential tRNA modification mutants in yeast, humans, mice and *C. elegans* (Zinshteyn et al., 2013; Nedialkova and Leidel, 2015; Laguesse et al., 2015; Thiaville et al., 2015; Chou et al., 2017; Tuorto et

al, 2018; Arango et al., 2018; Navarro et al., 2021 to name a few). In these experimental
systems codon-specific effects were consistent with the known modification targets using
long footprints.

In our opinion tRNA modification mutants are a better proxy to assess physiological
translation defects than cleaved tRNAs or uncharged tRNAs. Most single tRNA
modification mutants do not alter tRNA levels. Their tRNA are in most cases normally
charged but the modification defects reduce translation efficiency by modulating codon-
anticodon interactions of specific tRNAs. In the case of wobble uridine modification
mutants, the codon-specific translation defects have been independently verified by
orthogonal biophysical methods (Ranjan and Rodnina, 2017). Here, the precise step of
translational perturbations is known but not expected to induce empty A-sites.

In our experience the quality of codon-specific results during translation elongation
strongly depends on the amount of high-quality reads. A narrow cut of long footprints
following a strong digest largely avoids rRNA contamination and drastically increases
the numbers of high-quality reads thereby allowing for precise A-site mapping.

We highly appreciate the work of Wu *et al.* for their important contribution to expanding
the toolbox for ribosome profiling experiments. However, we disagree with the
interpretation that their study devaluates the analyses of 28 nt footprints. Based on the
reanalysis of the Wu *et al.* data, the analysis of 28 nt footprints in a large number of other
studies, and comparison of short footprints to long footprints in our data we conclude that
long footprints are relevant for addressing biological questions related to translation
dynamics including pausing.

We thank this reviewer for asking us to clarify this point. Otherwise, we would not have
analyzed our own and also the Wu *et al.*, data in such detail. We have gained important
insights to plan our own future experiments.

We mention these points now in our discussion.

Short footprints either derive from rotated ribosomes or from unrotated pre-
accommodation-state ribosomes with an empty A-site^{32,37}. In libraries that are devoid
of CHX short and long footprints simultaneously occur in yeast³², while both types of
footprints are present in human libraries independent of the CHX regimen (Fig. 1B
and Supplementary Fig. 1H-1M). While empty A-sites can be induced by treatments
like specific amino-acid starvation or the use of tRNA toxins³⁷, it is difficult to
distinguish these two types of short footprints *in vivo*. By combining CHX with other
inhibitors like tigecycline or anisomycin in yeast specific types of short footprints can
be enriched, allowing to further improve the resolution of ribosome profiling³⁷.
Nevertheless, the use of canonical long footprints and CHX have allowed to identify
biological meaningful features in ribosome profiling experiments e.g. in the analysis
of tRNA modification mutants^{8,39-41,43}. We have analyzed several features in ribosome
profiling data in response to CHX exposure independently for small and large
footprints and found the results to be very similar. While we cannot exclude that subtle
differences exist, these appear to be rather of quantitative than of qualitative nature.
Currently, it is unclear whether the simultaneous purification of both footprints sizes
increases the information gain in cases when inhibitor cocktails are not used. The
recovery of short and long footprints requires a wide size selection from the
acrylamide gel, thereby increasing the contamination with rRNA. Since the quality of
the results correlates with the amount of usable data an increase of contamination is a
factor that needs to be carefully balanced. Thus, unless the research question
necessitates the preparation of small footprints e.g., when empty A-sites are expected,
the use of large footprints appears fully justified. This strategy has the added

advantage of being able to compare the data to hundreds of published datasets that
have used long footprints. Nevertheless, the use of inhibitor cocktails has provided us
with new exiting strategies to further probe translation dynamics.

*3. The methods provided are inadequate. This is surprising given that a main point of the study*
*is that technical variation causes significant differences in the data. There is no information*
*about how the libraries were prepared after the isolation of monosomes from sucrose*
*gradients. What size markers were used to select footprint RNA? Without this information,*
*observations about the observed distributions of footprint lengths are meaningless.*

We agree that we have not been clear enough in our methods. We have therefore extended
the material and methods section and provide the full updated protocol as supplementary
information.

*4. The authors should repeat all of the codon-level analyses of Fig. 3 and Fig. 4 using the 21nt*
*footprints and determine the correlations to tAI, rare codons, etc.*

We have analyzed the codon occupancy of short footprints at the A-, P- and E-sites
(Fig. 3). We cannot provide an analysis of short footprints for most datasets included in
Fig. 4. However, we performed a wave analysis for short footprints using our own data
as a new Supplementary Fig. 4.

We refer to Fig. 3 in the results:

To test for the CHX-dependent enrichment of codons, we calculated transcriptome-
wide codon occupancy for the A-, P- and E-site codons in human and yeast (Fig. 3A-
3C, Supplementary Fig. 3A and data not shown). For short and long footprints,
correlations of A-site-codon occupancy in HEK 293T libraries were very high,
independent of the treatment ($R^2 \geq 0.92$ between all conditions; Fig. 3A). Similarly,
P-site-codon occupancy correlated well between treatments ($R^2 \geq 0.74$; Fig. 3B) while
CHX-pre-treatment changed E-site-codon occupancy markedly ($R^2 = 0.28-0.35$ for
$+/+$ and $-/-$ conditions; Fig. 3C).

We refer to Supplementary Fig. 4 in the results:

CGA exhibits a more pronounced wave than CGG and is read by $\text{tRNA}_{\text{ICG}}^{\text{Arg}}$ requiring
a wobble interaction to be decoded^{56,33}. As expected, we observed a wave for CGA
and CGG in the $+/+$ but not in the $-/+$ and $-/-$ libraries in yeast (Fig. 4A, and data not
shown). However, we did not observe a similar effect for CGA in HEK 293T cells
(Fig. 4B and Supplementary Fig. 4). In humans, decoding of CGA relies on its cognate
$\text{tRNA}_{\text{UCG}}^{\text{Arg}}$ and does not require wobble interactions^{54,55}. Furthermore, there are no
codons in humans that are decoded by a single tRNA like $\text{tRNA}_{\text{CCG}}^{\text{Arg}}$ in yeast
(Supplementary Table 1). Consistently, we did not observe wave formation for any
codon in humans (data not shown). Even for the UUA codon that is both rare and has
a low tRNA copy number in humans, we did not detect CHX-induced downstream
enrichment of ribosomes (Fig. 4B and Supplementary Fig. 4).

Fig. 3: Cycloheximide (CHX) does not alter mammalian ribosome occupancy at the A-site and the P-site. A) Correlation analysis of transcriptome-wide A-site codon occupancy in HEK 293T cells across different CHX treatments for short and long footprints (mean; n=3). Each black dot represents a codon. The size of the box indicates p-values. Correlations with a p-value > 0.05 are crossed out. B) Like A) for P-site codon occupancy. C) Like A) for E-site codon occupancy. Footprints were excised between 18-32 nt but are represented according to size: short footprints (21-22 nt) and long footprints (29-31 nt).

Supplementary Fig. 4: Cycloheximide (CHX) pre-treatment does not alter ribosome occupancy downstream of rare codons. Transcriptome-wide ribosome enrichment profiles surrounding CGA and UUA codons according to Hussmann *et al.*¹³ for short footprints (21 and 22 nt) of HEK 293T cells using different CHX treatment regimens.

We provide an analysis of tAI, codon usage and different amino acid properties for short and long footprints in human cells. tAI correlation of A-site occupancy is also provided for the HeLa data of Wu *et al.*, 2019 (Supplementary Fig. 5C). In humans, correlations with $1/tAI$ are not as good as in yeast. This is not surprising, since tRNA genes vary in their expression in mammals, which is not the case in yeast (Kutter *et al.*, 2011).

We discuss the data in the manuscript.

Recently, short footprints have entered the stage in addition to the canonical long footprints as a tool to obtain more information about ribosomal conformations^{32,37}. In yeast, short footprints show a high anti-correlation with the tAI. However, no significant correlation with $1/tAI$ is seen in human cells (Supplementary Fig. 5C). Similarly, the correlation between codon occupancy and amino acid polarity observed for short footprints in yeast³² is absent in human cells (Supplementary Fig. 5D; Hydrophobicity index). In fact, we did not find significant correlation between codon occupancy and most of the biochemical properties of amino acids for both long and short footprints (Supplementary Fig. 5D).

Supplementary Fig. 5C: Spearman rank correlation of A-site codon occupancy with tRNA adaptation index for HeLa⁴ and HEK 293T cells for short and long footprints.

The size of the box indicates p-values. Correlations with a p-value > 0.05 are crossed out.

Supplementary Fig. 5D: Spearman rank correlation of A-, P- and E-sites codon occupancy with various amino acid properties and codon usage in HEK 293T cells for short and long footprints. The size of the box indicates p-values. Correlations with a p-value > 0.05 are crossed out.

Additional points:

5. It is not clear what justifies the strongly worded conclusion that “Cycloheximide does not affect the translation ramp” (Fig. 2). The normalized coverage profiles in Fig. 2A are strikingly different dependent on treatment (+/+, -/+, -/- cycloheximide).

This might be a misunderstanding. Our use of the term "translational ramp" follows Tuller *et al.*, 2010, Gerashchenko *et al.*, 2014 and Weinberg *et al.*, 2016. There the ramp describes the beginning of transcripts between roughly 15 to 200 codons. As rightfully pointed out by this reviewer, ribosome coverage at start codons differs clearly between the +/+ and -/+ or -/- libraries. This is because pre-incubation using standard CHX concentrations (100 µg/ml) blocks elongation but not initiation. This will affect the number of reads at the initiation site as a function of initiation rates and the speed of

923 sample processing. To not bias our analysis, we, therefore, excluded the first 15 codons
from most analyses.

We more specifically speak about the increase of read number around the start codon in
response to CHX pre-treatment in the results:

Several studies in yeast have described an increase in ribosome density in the first
~200 codons of ORFs. This phenomenon is called the '5' translation ramp' and is
thought to be caused by a slow elongation speed in this region^{7,44,45}. In contrast, a
study in murine embryonic stem cells concluded that a similar ramp does not occur in
vertebrate cells⁵. To elucidate the effect of CHX on this phenomenon, we analyzed
footprint density in the beginning of the open reading frames. Importantly, we found
an enrichment of ribosomes from 15 to ~150 codons irrespective of drug treatment
and footprint length in human cells (Fig. 2A and Supplementary Fig. 2A, 2B and 2C).
The scale of this effect and the position of the maximum footprint density differ
between human and yeast (~60 codons in humans, and ~40 codons in yeast;
Supplementary Fig. 2D). However, the ramp is independent of gene-translation levels
and gene length (Supplementary Fig. 2B, 2C, 2E and data not shown). This shows that
the translation ramp itself is not induced by CHX but is a genuine biological feature
of translation in humans as well as in yeast. Multiple factors were proposed to trigger
the translational ramp, such as the presence of rare codons with low cognate tRNA
availability, mRNA secondary structures or the interaction of nascent polypeptides
with the ribosomal exit tunnel^{39,40}. Our results exclude that CHX treatment, gene
length or transcript expression levels are primary contributing factors (Fig. 2 and
Supplementary Fig. 2). However, ribosome density is increased in the first ~15 codons
in human and yeast +/+ libraries and to a lesser extent in the -/+ and -/- libraries (Fig.
2A and Supplementary Fig. 2A and 2B). This is consistent with the observation that
CHX arrests elongation but does not block translation initiation at the concentration
generally used in ribosome profiling²⁹. Therefore, we do not recommend to pre-treat
cultures with CHX. In case that an experiment requires CHX pre-incubation, time
should be kept to a minimum. Furthermore, the first codons should be excluded from
gene-expression analyses since this accumulation will skew the analysis of differential
gene translation towards initiation rates and does not reflect global gene translation
levels.

And in the discussion:

Second, CHX blocks elongation but not initiation. This leads to an enrichment of reads
around the initiation site as a function of initiation rate and the extent of CHX
treatment. While this effect can be used intentionally to enrich for initiation sites in
species where harringtonine and lactimidomycin do not work, it is generally advised
to exclude the first few codons of a gene from most analyses.

*6. Fig. 2A shows a >2-fold accumulation of ribosomes at stop codons in the (-/+) sample*
*compared to the others. This observation strongly suggests there is significant ribosome*
*movement through the CDS in the absence of CHX added to the cells. It is not clear which of*
*the treatments produces the best approximation of the desired 'snapshot' of ribosome locations*
*in vivo.*

We thank this reviewer for pointing to this. Indeed, it looks like a fraction of ribosomes
moves in the context of the stop codon in -/+ libraries. However, it is unclear which of

the three conditions reflects the *in vivo* situation most adequately. We expect to see
increased coverage at the stop codon since termination takes more time than individual
elongation cycles. This is the case in all three treatment regimens.

Cycloheximide has been shown to stabilize ribosomal termination complexes *in vitro*
likely by locking them in an unrotated state (Susorov et al., 2015). If the release of
ribosomes is blocked by CHX, this might explain the observed differences: First, in +/+
conditions ribosomes fully arrest on mRNA. Therefore, they do not accumulate at stop
codons and do not terminate. Second, in -/+ conditions ribosomes might move for one or
two additional cycles before being captured by CHX thereby increasing the occupancy
at the stop codon. However, since CHX blocks their release the read number at stop
codons increases. Third, in -/- conditions ribosomes will move into the stop codons in
the extract. However, termination is not blocked by CHX. Therefore, the release can
occur leading to a lower number of stop codon reads when compared to the -/+
conditions.

Since our data does not allow us to reach a conclusion about which condition reflects the
*in vivo* situation best, we briefly discuss this point. To really address this question, it
would be needed to combine the different treatments with mutants of release factors that
block termination at specific steps or even use combinations of inhibitors. This however
goes beyond the scope of this article. Nevertheless, we wanted to report the differences
such that researchers who are interested in analyzing termination are aware of this.

We discuss this point in the results:

Finally, we observed an accumulation of ribosomal footprints at stop codons in all
three conditions, but most prominently in the -/+ libraries (Fig. 2A and Supplementary
Fig. 2A and 2D), consistent with reports in murine and yeast cells^{5,33,45}. This
emphasizes that the steps during translation termination differ from normal translation
elongation and is consistent with the observation that termination complexes are
differentially stabilized by the addition of CHX⁴⁸. It is possible that ribosome splitting
is affected by CHX and does not occur in the -/+ extracts while it can still occur in
untreated -/- samples. While it is yet unclear, which of the three treatment options
captures the *in vivo* situation at the stop codon most realistically, this effect can likely
be exploited to complement *in vitro* termination experiments.

And in the discussion:

Third, termination differs from normal elongation cycles. Ribosomes spend more time
at stop codons, which is apparent in all treatment conditions. However, the strongest
enrichment is seen in -/+ libraries. Ribosomes are likely able to enter the termination
cycle but fail to terminate since CHX in the lysis buffer locks the ribosomes in the
unrotated state thereby stabilizing termination complexes⁴⁸. To specifically analyze
termination, it might be beneficial to use the different treatments side by side.

**References:**

Arango, Daniel, David Sturgill, Najwa Alhusaini, Allissa A. Dillman, Thomas J. Sweet, Gavin
Hanson, Masaki Hosogane, et al. 2018. Acetylation of Cytidine in mRNA Promotes
Translation Efficiency. *Cell* 175 (7): 1872-1886.e24.

Chan, Patricia P., and Todd M. Lowe. 2016. GtRNADB 2.0: An Expanded Database of Transfer
RNA Genes Identified in Complete and Draft Genomes. *Nucleic Acids Research* 44 (D1):
D184–89.

Chou, Hsin-Jung, Elisa Donnard, H. Tobias Gustafsson, Manuel Garber, and Oliver J. Rando.
2017. Transcriptome-Wide Analysis of Roles for tRNA Modifications in Translational
Regulation. *Molecular Cell* 68 (5): 978-992.e4.

Gerashchenko, Maxim V., and Vadim N. Gladyshev. 2014. Translation Inhibitors Cause
Abnormalities in Ribosome Profiling Experiments. *Nucleic Acids Research* 42 (17): e134–
e134.

Hussmann, Jeffrey A., Stephanie Patchett, Arlen Johnson, Sara Sawyer, and William H. Press.
2015. Understanding Biases in Ribosome Profiling Experiments Reveals Signatures of
Translation Dynamics in Yeast. Edited by Michael Snyder. *PLOS Genetics* 11 (12):
e1005732.

Kearse, Michael G., Daniel H. Goldman, Jiou Choi, Chike Nwaezeapu, Dongming Liang,
Katelyn M. Green, Aaron C. Goldstrohm, Peter K. Todd, Rachel Green, and Jeremy E.
Wilusz. 2019. Ribosome Queuing Enables Non-AUG Translation to Be Resistant to Multiple
Protein Synthesis Inhibitors. *Genes & Development* 33 (13–14): 871–85.

Kutter, Claudia, Gordon D Brown, Ângela Gonçalves, Michael D Wilson, Stephen Watt, Alvis
Brazma, Robert J White, and Duncan T Odom. 2011. “ol III Binding in Six Mammals Shows
Conservation among Amino Acid Isoforms despite Divergence among tRNA Genes. *Nature*
*Genetics* 43 (10): 10.

Laguesse, Sophie, Catherine Creppe, Danny D. Nedialkova, Pierre-Paul Prévot, Laurence
Borgs, Sandra Huysseune, Bénédicte Franco, et al. 2015. A Dynamic Unfolded Protein
Response Contributes to the Control of Cortical Neurogenesis. *Developmental Cell* 35 (5):
553–67.

Lareau, Liana F, Dustin H Hite, Gregory J Hogan, and Patrick O Brown. 2014. Distinct Stages
of the Translation Elongation Cycle Revealed by Sequencing Ribosome-Protected mRNA
Fragments. *ELife* 3 (May): e01257.

Lawless, Craig, Richard D Pearson, Julian N Selley, Julia B Smirnova, Christopher M Grant,
Mark P Ashe, Graham D Pavitt, and Simon J Hubbard. 2009. Upstream Sequence Elements
Direct Post-Transcriptional Regulation of Gene Expression under Stress Conditions in Yeast.
*BMC Genomics* 10 (1): 7.

Letzring, D. P., K. M. Dean, and E. J. Grayhack. 2010. Control of Translation Efficiency in
Yeast by Codon-Anticodon Interactions. *RNA* 16 (12): 2516–28.

Letzring, D. P., A. S. Wolf, C. E. Brule, and E. J. Grayhack. 2013. Translation of CGA Codon
Repeats in Yeast Involves Quality Control Components and Ribosomal Protein L1. *RNA* 19
(9): 1208–17.

Navarro, Isabela Cunha, Francesca Tuorto, David Jordan, Carine Legrand, Jonathan Price,
Fabian Braukmann, Alan G Hendrick, et al. 2021. Translational Adaptation to Heat Stress Is
Mediated by RNA 5-methylcytosine in *Caenorhabditis Elegans*. *The EMBO Journal* 40 (6).

Nedialkova, Danny D., and Sebastian A. Leidel. 2015. Optimization of Codon Translation
Rates via tRNA Modifications Maintains Proteome Integrity. *Cell* 161 (7): 1606–18.

Ranjan, Namit, and Marina V. Rodnina. 2017. Thio-Modification of TRNA at the Wobble
Position as Regulator of the Kinetics of Decoding and Translocation on the Ribosome.
*Journal of the American Chemical Society* 139 (16): 5857–64.

Reis, Mario dos, Renos Savva, and Lorenz Wernisch. 2004. Solving the Riddle of Codon Usage
Preferences: A Test for Translational Selection. *Nucleic Acids Research* 32 (17): 5036–44.

Santos, Daniel A, Lei Shi, Benjamin P Tu, and Jonathan S Weissman. 2019. Cycloheximide
Can Distort Measurements of mRNA Levels and Translation Efficiency. *Nucleic Acids*
*Research* 47 (10): 4974–85.

Thiaville, Patrick, Rachel Legendre, Diego Rojas-Benitez, Agnes Baudin-Baillieu, Isabelle
Hatin, Guilhem Chalancon, Alvaro Glavic, Olivier Namy, and Valerie de Crecy-Lagard.
2016. Global Translational Impacts of the Loss of the tRNA Modification t⁶A in Yeast.
*Microbial Cell* 3 (1): 29–45.

Tuller, Tamir, Asaf Carmi, Kalin Vestsigian, Sivan Navon, Yuval Dorfan, John Zaborske, Tao
Pan, Orna Dahan, Itay Furman, and Yitzhak Pilpel. 2010. An Evolutionarily Conserved
Mechanism for Controlling the Efficiency of Protein Translation. *Cell* 141 (2): 344–54.

Tuorto, Francesca, Carine Legrand, Cansu Cirzi, Giuseppina Federico, Reinhard Liebers,
Martin Müller, Ann E Ehrenhofer-Murray, Gunnar Dittmar, Hermann-Josef Gröne, and Frank
Lyko. 2018. Queuosine-modified TRNAs Confer Nutritional Control of Protein Translation.
*The EMBO Journal* 37 (18).

Urban, Jörg, Alexandre Soulard, Alexandre Huber, Soyeon Lippman, Debdyuti
Mukhopadhyay, Olivier Deloche, Valeria Wanke, et al. 2007. Sch9 Is a Major Target of
TORC1 in *Saccharomyces Cerevisiae*. *Molecular Cell* 26 (5): 663–74.

Weinberg, David E., Premal Shah, Stephen W. Eichhorn, Jeffrey A. Hussmann, Joshua B.
Plotkin, and David P. Bartel. 2016. Improved Ribosome-Footprint and mRNA Measurements
Provide Insights into Dynamics and Regulation of Yeast Translation. *Cell Reports* 14 (7):
1787–99.

Wu, C. C. C., B. Zinshteyn, K. A. Wehner, and R. Green. 2019. High-Resolution Ribosome
Profiling Defines Discrete Ribosome Elongation States and Translational Regulation during
Cellular Stress. *Molecular Cell* 73 (5): 959-970.e5.

Zinshteyn, Boris, and Wendy V. Gilbert. 2013. Loss of a Conserved TRNA Anticodon
Modification Perturbs Cellular Signaling. Edited by Gregory P. Copenhaver. *PLoS Genetics*
9 (8): e1003675.

Reviewers' Comments:

Reviewer #2:

Remarks to the Author:

The authors have satisfactorily addressed my concerns with their revisions:

Points 1,2, and 4: I greatly appreciate their inclusion of short footprint analysis (of their own data) and thorough reanalysis of the Wu et al. ribosome profiling data. Although I am not certain that the quantitative dampening of pause signal in the 28mers compared to 21mers (Supp. Fig. 10, p.18 of rebuttal) is unimportant, I am persuaded by the authors' argument that "the quality of codon-specific results during translation elongation [analysis by ribosome profiling] strongly depends on the amount of high-quality reads." Thus, even if short footprints are more sensitive to reveal some kinds of ribosome pause sites, libraries prepared by cutting narrowly around 28nts may nevertheless provide better evidence for these pauses.

Point 3: The revised methods contain sufficient details for other investigators to follow.

**Reviewer #2 (Remarks to the Author):**

*The authors have satisfactorily addressed my concerns with their revisions:*

*Points 1,2, and 4: I greatly appreciate their inclusion of short footprint analysis (of their own*
*data) and thorough reanalysis of the Wu et al. ribosome profiling data. Although I am not*
*certain that the quantitative dampening of pause signal in the 28mers compared to 21mers*
*(Supp. Fig. 10, p.18 of rebuttal) is unimportant, I am persuaded by the authors' argument*
*that "the quality of codon-specific results during translation elongation [analysis by*
*ribosome profiling] strongly depends on the amount of high-quality reads." Thus, even if*
*short footprints are more sensitive to reveal some kinds of ribosome pause sites, libraries*
*prepared by cutting narrowly around 28nts may nevertheless provide better evidence for*
*these pauses.*

*Point 3: The revised methods contain sufficient details for other investigators to follow.*

We are pleased to see that we have been able to address all concerns raised by this
reviewer and thank him/her for the positive assessment of our revised manuscript.